# CMG helicase disassembly is controlled by replication fork DNA, replisome components and a ubiquitin threshold

Tom D Deegan*, Progya P Mukherjee[†], Ryo Fujisawa[†], Cristian Polo Rivera[†], Karim Labib*

The MRC Protein Phosphorylation and Ubiquitylation Unit, School of Life Sciences, University of Dundee, Dundee, United Kingdom

**Abstract** The eukaryotic replisome assembles around the CMG helicase, which stably associates with DNA replication forks throughout elongation. When replication terminates, CMG is ubiquitylated on its Mcm7 subunit and disassembled by the Cdc48/p97 ATPase. Until now, the regulation that restricts CMG ubiquitylation to termination was unknown, as was the mechanism of disassembly. By reconstituting these processes with purified budding yeast proteins, we show that ubiquitylation is tightly repressed throughout elongation by the Y-shaped DNA structure of replication forks. Termination removes the repressive DNA structure, whereupon long K48-linked ubiquitin chains are conjugated to CMG-Mcm7, dependent on multiple replisome components that bind to the ubiquitin ligase SCF[Dia2]. This mechanism pushes CMG beyond a '5-ubiquitin threshold' that is inherent to Cdc48, which specifically unfolds ubiquitylated Mcm7 and thereby disassembles CMG. These findings explain the exquisite regulation of CMG disassembly and provide a general model for the disassembly of ubiquitylated protein complexes by Cdc48.

*For correspondence:
tdeegan@dundee.ac.uk (TDD);
kpmlabib@dundee.ac.uk (KL)

[†]These authors contributed equally to this work

Competing interests: The authors declare that no competing interests exist.

## Introduction

Eukaryotic chromosomes are duplicated just once per cell cycle, by a large molecular machine called the replisome (*Bell and Labib, 2016*; *Burgers and Kunkel, 2017*). Replisome assembly is initiated in the G1-phase of the cell cycle, when the six Mcm2-7 ATPases are loaded around double strand DNA (dsDNA) at origins of replication, as a 'head-to-head' double hexamer (*Evrin et al., 2009*; *Li et al., 2015*; *Remus et al., 2009*). Each Mcm2-7 hexamer comprises a two-tiered ring, which represents the catalytic core of the helicase that subsequently unwinds the DNA duplex at replication forks (*Aparicio et al., 1997*; *Ishimi, 1997*; *Labib et al., 2000*).

Origin unwinding cannot occur until S-phase, when two protein kinases induce multiple 'firing factors' to recruit the remaining helicase subunits known as Cdc45 and the GINS complex. The association of Mcm2-7 with Cdc45 and GINS splits the Mcm2-7 double hexamer into two CMG (Cdc45-Mcm2-7-GINS) helicase complexes (*Abid Ali et al., 2016*; *Douglas et al., 2018*; *Moyer et al., 2006*). CMG assembly is coupled to initial origin melting (*Douglas et al., 2018*), so that each of the two nascent CMG helicases is associated with around 6–7 bp of unwound single-strand DNA (ssDNA). Finally, the transient opening of a poorly defined 'gate' in CMG leads to complete exclusion of the lagging strand DNA template from the Mcm2-7 central channel (*Wasserman et al., 2019*), in a step that requires the Mcm10 protein (*Douglas et al., 2018*). This produces two activated CMG helicases that encircle opposite strands of the parental DNA duplex. The two helicases then bypass each other and begin to unwind the parental duplex DNA in opposite directions, led by the N-terminal tier of Mcm2-7 and driven by the ATPases in the Mcm2-7 C-terminal tier (*Douglas et al., 2018*; *Georgescu et al., 2017*). Multiple factors associate with the two helicases to form two replisomes (*Gambus et al., 2006*; *Sengupta et al., 2013*), leading to the initiation of DNA

synthesis at a pair of bidirectional replication forks, with CMG encircling the template of the leading strand.

The association of CMG with replication fork DNA is uninterrupted from initiation to termination, based on the stable entrapment of the leading strand DNA template in the central channel of Mcm2-7. This is important, because displacement of CMG from DNA blocks fork progression irreversibly (*Labib et al., 2000*), since CMG cannot be reassembled at a replication fork during S-phase. Nonetheless, the convergence of two replication forks leads very quickly to CMG disassembly and replisome dissolution (*Dewar and Walter, 2017*; *Gambus, 2017*), which is likely to represent the final stage of DNA replication termination. Consistent with this view, CMG unloading occurs after the formation of a fully ligated DNA product during plasmid replication in *Xenopus laevis* egg extracts (*Dewar et al., 2015*). Moreover, we observed complete plasmid replication and the formation of covalently closed products, using a reconstituted yeast replication system that lacks the components required for CMG disassembly (*Deegan et al., 2019*). This indicated that post-termination CMG complexes must encircle dsDNA, after DNA synthesis has been completed.

The key regulated step during replisome disassembly is the ubiquitylation of the Mcm7 subunit of CMG (*Maric et al., 2014*; *Moreno et al., 2014*), which in budding yeast is mediated by the cullin ubiquitin ligase known as SCF$^{Dia2}$ and the ubiquitin conjugating enzyme Cdc34 (*Maric et al., 2014*). Until now, it was unknown how CMG ubiquitylation is robustly blocked throughout initiation and elongation and then induced with high efficiency during termination.

A termination-specific signal that triggers CMG ubiquitylation has not been identified (*Dewar and Walter, 2017*), but several possibilities have been proposed (*Figure 1—figure supplement 1*). Firstly, ubiquitylation might be triggered by the presence of dsDNA within the central channel of the Mcm2-7 component of CMG (*Dewar et al., 2015*), which occurs after fork convergence, once each CMG helicase encounters the 5' end of the opposing fork's nascent lagging strand. The presence of dsDNA in the Mcm2-7 channel was suggested to induce a termination-specific conformational change in CMG that leads to recruitment or activation of the cullin ubiquitin ligase (*Dewar and Walter, 2017*; *Maric et al., 2014*). Alternatively, CMG ubiquitylation might be dependent upon the juxtaposition of two converged replisomes (*Dewar and Walter, 2017*), for example if the ligase could only be recruited to a pair of converged replisomes, or if ubiquitylation can only occur in trans, or else if convergence drives dimerisation of a replisome-associated ubiquitin ligase, as reported previously for another SCF enzyme (*Tang et al., 2007*). Finally, it was suggested that ubiquitylation might be triggered by the encounter of CMG on dsDNA with the rear face of PCNA complexes from the converging fork (*Dewar and Walter, 2017*). Current data do not distinguish between these or other possibilities. Importantly, all the above models are dependent on the convergence of two replication forks, and thus provide no insight into the fate of CMG when replication is terminated in other ways, such as when a single fork meets a telomere, or when a fork meets a nick in the leading strand DNA template. CMG is likely to slide off DNA under such conditions, and its subsequent fate is important, given recent data showing that Mcm10 can mediate the reloading of CMG onto ssDNA in vitro (*Wasserman et al., 2019*).

Following ubiquitylation, CMG is disassembled by Cdc48/p97 (*Dewar et al., 2017*; *Maric et al., 2014*; *Moreno et al., 2014*; *Sonneville et al., 2017*), which is a hexameric ATPase that disrupts protein structure and transports unfolded polypeptides through its central channel (*Blythe et al., 2017*; *Bodnar and Rapoport, 2017*; *Cooney et al., 2019*; *Weith et al., 2018*). Cdc48 is recruited to ubiquitylated substrates by its Ufd1-Npl4 cofactors (*Blythe et al., 2017*; *Bodnar and Rapoport, 2017*; *Tsuchiya et al., 2017*). These have been found to bind with varying affinity to K48-linked chains of three or more ubiquitins (*Bodnar and Rapoport, 2017*; *Twomey et al., 2019*), or of at least six ubiquitins (*Tsuchiya et al., 2017*). However, the functional link between ubiquitin chain length and substrate unfolding by Cdc48 has yet to be examined directly. Previous studies have not determined the mechanism by which Cdc48 disassembles ubiquitylated protein complexes such as CMG. Recruitment of Cdc48 to non-ubiquitylated proteins can induce unfolding (*Cheng and Chen, 2015*; *Wang and Ye, 2018*; *Weith et al., 2018*), and so it remains unclear which subunits in a ubiquitylated protein complex would be targeted by Cdc48-Ufd1-Npl4.

Here, we address the mechanism and regulation of CMG ubiquitylation and disassembly, by recapitulating these reactions in vitro with purified budding yeast proteins.

# Results

## DNA-dependent repression of CMG ubiquitylation during initiation and elongation

Previous studies described the reconstitution of DNA replication initiation and replisome assembly with yeast proteins, leading to the progression of bi-directional replication forks away from an origin (*Yeeles et al., 2015*; *Yeeles et al., 2017*). However, DNA replication termination is defective when two forks converge in this system, unless the reactions also contain one of the two yeast members of the Pif1 helicase family, which help the replisome to unwind the final stretch of parental DNA (*Deegan et al., 2019*). For this reason, replication reactions in the presence or absence of Pif1 provide a model system, with which to study the regulation of CMG ubiquitylation during DNA replication termination.

Therefore, we replicated a 3.2 kb plasmid plus or minus Pif1, before adding purified SCF$^{Dia2}$ together with the other necessary components of the ubiquitylation system (*Figure 1A*, *Figure 2A*). In the absence of Pif1, converging replication forks stalled to produce a 'late replication intermediate' (*Figure 1B*, lane 2). Under these conditions, CMG-Mcm7 ubiquitylation was not observed (*Figure 1C*, lane 2). However, Mcm7 ubiquitylation was readily detected in reactions containing Pif1 (*Figure 1C*, lane 1), which supported DNA replication termination and the production of fully replicated 3.2 kb plasmids (*Figure 1B*, lane 1). Moreover, Mcm7 ubiquitylation was dependent upon Pif1 helicase activity (*Figure 1D* (ii), compare lanes 2 and 3; Pif1-K264A lacks helicase activity and cannot support DNA replication termination). These findings indicated that CMG ubiquitylation is tightly blocked before DNA replication termination in the reconstituted replication system, without requiring a deubiquitylase enzyme to counteract SCF$^{Dia2}$ and the ubiquitin conjugation system. The data also demonstrate that CMG-Mcm7 ubiquitylation is induced when converging replication forks undergo DNA replication termination, without needing the action of any additional factors other than the E1, E2 and E3 ubiquitylation enzymes.

As discussed above, the initiation reaction at origins of DNA replication generates two converged CMG helicases that must bypass each other (*Champasa et al., 2019*; *Douglas et al., 2018*; *Georgescu et al., 2017*). Nevertheless, nascent CMG helicases are not immediately disassembled during initiation in yeast cells (*Gambus et al., 2006*; *Kanemaki and Labib, 2006*), nor does initiation lead to CMG-Mcm7 ubiquitylation in the reconstituted replication system, even when SCF$^{Dia2}$, E1 and E2 are present from the start of the reaction (*Figure 1D* (ii), lane 1). Therefore, it is unlikely that the juxtaposition of two converged CMG complexes represents a trigger for CMG ubiquitylation, either during initiation or when forks converge during DNA replication termination (*Figure 1—figure supplement 1*).

In the absence of Mcm10, the initial steps of initiation convert an Mcm2-7 double hexamer into two CMG helicases that are each associated with around 6–7 bp of unwound DNA (*Douglas et al., 2018*), indicating that the lagging strand template has not yet been fully excluded from the Mcm2-7 channel of CMG. The newly assembled CMG helicases persist and remain associated with origins in yeast cells that lack Mcm10 (*Kanke et al., 2012*; *van Deursen et al., 2012*; *Watase et al., 2012*). Correspondingly, CMG ubiquitylation is blocked in the reconstituted in vitro replication system, when replication initiates in the absence of Mcm10 (*Figure 1E* (i), compare lanes 1–2).

The failure to ubiquitylate CMG under such conditions might simply reflect the absence of an activating signal that is specific to termination. Alternatively, however, it is possible that ubiquitylation of nascent CMG is actively repressed at origins, for example by the manner in which newly assembled CMG embraces origin DNA. To test this idea, we performed replication reactions in the presence or absence of Mcm10, and then digested the products with DNase, before adding SCF$^{Dia2}$ and other ubiquitylation enzymes. Remarkably, both nascent CMG and post-termination CMG were ubiquitylated to an equivalent degree upon release from DNA (*Figure 1E* (i), compare lanes 3–4; the efficiency of DNase digestion is shown in *Figure 1—figure supplement 2*). These findings strongly indicate that the regulation of CMG ubiquitylation during chromosome duplication is not based on an activating signal that can only occur during DNA replication termination at a pair of converged forks. Instead, nascent CMG is potentially a substrate for SCF$^{Dia2}$, but ubiquitylation is blocked during initiation, dependent upon the DNA structure of the origin.

To test whether CMG ubiquitylation is also repressed in a DNA-dependent manner during elongation, we performed replication reactions in the absence or presence of Pif1 as above, and then

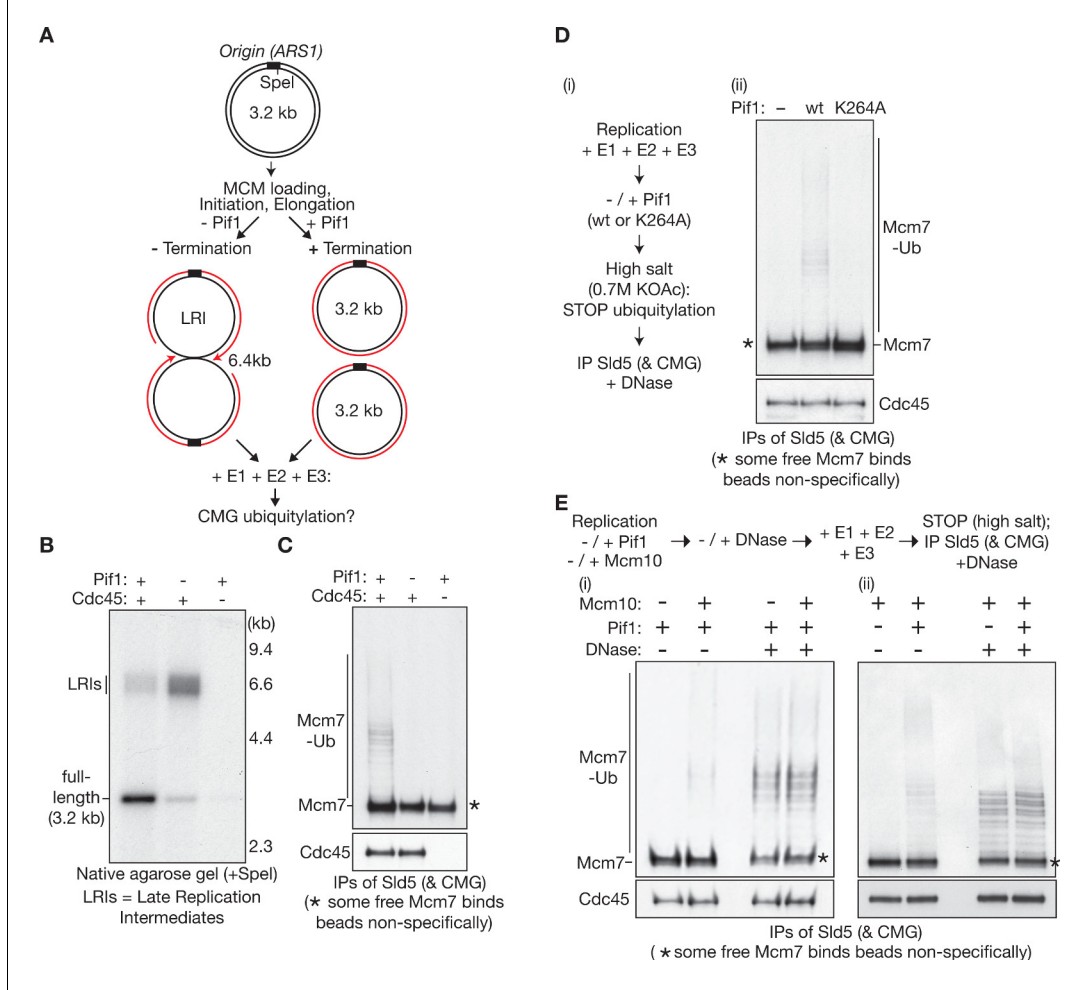

**Figure 1.** Mcm7 ubiquitylation is repressed before termination by the association of CMG with DNA. (**A**) Experimental scheme for B-C, based on in vitro replication of plasmid DNA with purified budding yeast proteins. LRI = Late Replication Intermediate. (**B**) Nascent DNA replication products were digested with SpeI and analysed by native agarose gel. (**C**) At the end of the reactions, the CMG helicase was released from DNA by treatment with DNase, in the presence of high salt to block further ubiquitylation, before isolation of CMG by immunoprecipitation of Sld5. The asterisk (*) indicates unmodified Mcm7, which binds non-specifically to beads under these conditions. (**D**) (**i**) Experimental scheme (Pif1 K264A is inactive as a helicase and does not support replication termination); (**ii**) The indicated CMG subunits were monitored by immunoblotting. (**E**) (**i**) Replication reactions were performed in the presence or absence of Mcm10 or Pif1, before treatment for 10' at 30°C with DNase to release CMG from DNA. Subsequently, the samples were incubated for 20 min in the presence of E1-E2-E3, and the reactions were then stopped by addition of high salt, before isolation of CMG as above (in the presence of DNase). The indicated subunits of CMG were monitored by immunoblotting. See also *Figure 1—figure supplements 1–2*.

The online version of this article includes the following figure supplement(s) for figure 1:

**Figure supplement 1.** Previous models for the regulation of CMG ubiquitylation during DNA replication termination.

**Figure supplement 2.** Efficient DNA digestion at end of in vitro replication reactions.

digested the products with DNase before addition of SCF^Dia2, E1 and E2. Strikingly, CMG released from pre-termination forks (*Figure 1E* (ii), lane 3, -Pif1) was ubiquitylated just as efficiently as CMG derived from fully replicated plasmid DNA (*Figure 1E* (ii), lane 4, +Pif1). Overall, therefore, these data indicate that the regulation of CMG ubiquitylation during chromosome replication is based on DNA-dependent repression during initiation and elongation, which is then relieved during DNA replication termination.

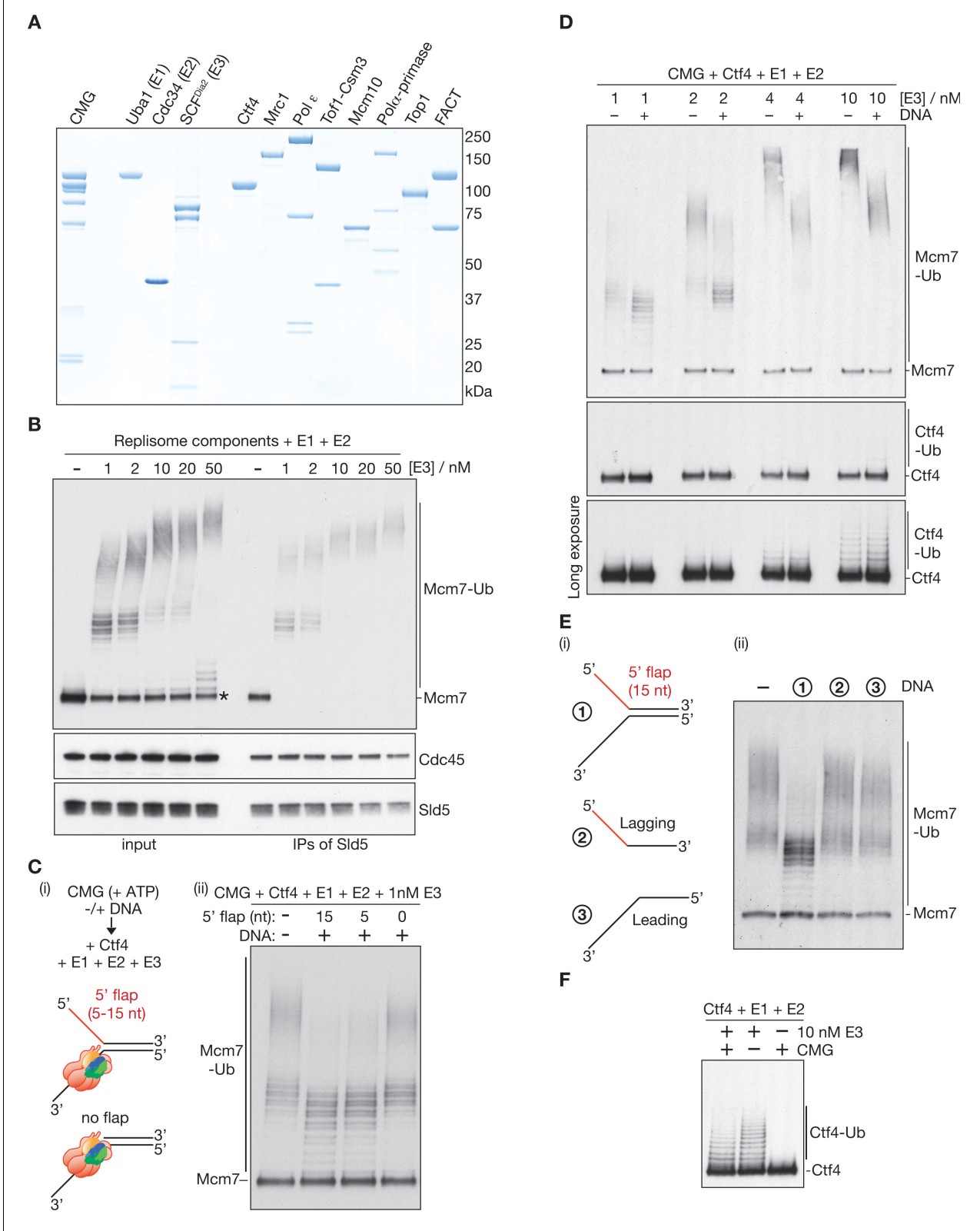

**Figure 2.** Replication fork structure inhibits CMG ubiquitylation by SCF[Dia2] and Cdc34. (**A**) Purified proteins used in this study, analysed by SDS-PAGE and stained with colloidal Coomassie Blue. (**B**) The components described in (**A**) were incubated for 20' at 30℃ with the indicated concentrations of E3 (input), before isolation of CMG (IPs of Sld5) and immunoblotting of the indicated components. The asterisk indicates a small pool of free Mcm7 that was not targeted for ubiquitylation. (**C**) (**i**) Reaction scheme and illustration of the association of CMG with model DNA replication forks. Replisome

*Figure 2 continued on next page*

*Figure 2 continued*

components other than Ctf4 were omitted to limit DNA unwinding by CMG; (ii) Ubiquitylation reactions in the presence or absence of synthetic replication forks with the indicated 5' flaps. (D) Reactions at the indicated [E3], plus or minus the model fork substrate with 15 nt 5' flap from (C). (E) Reactions were performed as in (C), with addition of the indicated DNA (2 and 3 are the ssDNA oligos that were used to make 1). (F) Ubiquitylation reactions performed as in (D), in the absence of DNA and +/- E3 or CMG as indicated. See also *Figure 2—figure supplements 1–2*.

The online version of this article includes the following figure supplement(s) for figure 2:

**Figure supplement 1.** Reconstituted CMG ubiquitylation involves the conjugation of K48-linked ubiquitin chains to Mcm7.

**Figure supplement 2.** Association of the CMG helicase with model DNA replication forks.

## CMG ubiquitylation is inherently efficient in the presence of other replisome proteins

To test directly whether CMG is an efficient substrate of SCF$^{Dia2}$ when not bound to DNA, we purified recombinant yeast CMG helicase as described previously (*Zhou et al., 2017*) and then incubated it together with the other replisome factors and ubiquitylation components described above (*Figure 2A–B*). Strikingly, even 1 nM of SCF$^{Dia2}$ was sufficient to ensure that 15 nM CMG was ubiquitylated with extremely high efficiency in the presence of other replisome factors (*Figure 2B*, right half; *Figure 2—figure supplement 1A* shows that the reaction was dependent upon addition of E1, E2 and E3). The reaction was highly selective for CMG-Mcm7 (*Figure 2—figure supplement 1B–C*) and involved the conjugation of long K48-linked ubiquitin chains onto more than one lysine of Mcm7 (*Figure 2—figure supplement 1A*, right panel). In contrast, ubiquitylation of the previously reported substrates Ctf4 and Mrc1 (*Mimura et al., 2009*) was undetectable at 1 nM SCF$^{Dia2}$ (*Figure 2—figure supplement 1D*), and instead required concentrations of SCF$^{Dia2}$ that are saturating for CMG ubiquitylation (*Figure 2—figure supplement 1E*). Overall, these findings demonstrated that SCF$^{Dia2}$ ubiquitylates CMG-Mcm7 with remarkable efficiency and selectivity, without any requirement for CMG to have passed through the steps of DNA replication termination. Moreover, these data show that CMG ubiquitylation does not require the presence of dsDNA in the central channel of Mcm2-7, or the encounter of CMG with the rear face of PCNA from a converged replication fork.

## The lagging strand template of a DNA replication fork impairs CMG ubiquitylation

The efficient ubiquitylation of recombinant CMG in a soluble system made it possible to explore how ubiquitylation might be regulated by model DNA substrates with defined structures. These all contained ssDNA that is equivalent to the leading strand template along which CMG tracks, together with variable lengths of ssDNA corresponding to the lagging strand template that is excluded from the Mcm2-7 channel of the helicase (*Figure 2C* (i); the two ssDNA regions were non-complementary, to prevent reannealing). The reactions included Ctf4 to stimulate CMG ubiquitylation (as discussed below), but other replisome components were omitted, in order to limit the rate of DNA unwinding by CMG.

Compared to reactions lacking DNA (*Figure 2C* (ii), lane 1), CMG-Mcm7 ubiquitylation was markedly impaired by an excess of fork DNA containing a short 5' flap that corresponded to the excluded DNA strand (*Figure 2C* (ii), lanes 2–3). Moreover, the inhibitory effect of fork DNA was observed over a broad range of E3 concentrations (*Figure 2D*). Note that free CMG, which is a highly efficient substrate for SCF$^{Dia2}$, is in equilibrium with DNA-bound CMG under such conditions. Therefore, any inhibition of CMG ubiquitylation that requires binding to fork DNA will only be partial in the soluble system, compared to the plasmid replication system where CMG is topologically trapped on its DNA template.

CMG ubiquitylation was not inhibited by an otherwise identical DNA substrate that lacked the excluded DNA strand (*Figure 2C* (ii), lane 4). Importantly, CMG associated equally well with all of the tested substrates (*Figure 2—figure supplement 2*). Moreover, ssDNA versions of either strand did not impair CMG ubiquitylation (*Figure 2E*). These findings indicated that the Y-shaped DNA structure of a replication fork inhibits the ability of SCF$^{Dia2}$ and Cdc34 to support CMG ubiquitylation.

Finally, we investigated whether the inhibitory effect of fork DNA was specific to CMG. As noted above, Ctf4 is a much less efficient substrate of SCF$^{Dia2}$ than CMG (*Figure 2D*, compare Ctf4 and

Mcm7), but Ctf4 ubiquitylation was detectable in the presence of 10 nM E3 ligase and was independent of CMG (*Figure 2F*). Notably, Ctf4 ubiquitylation under these conditions was not affected by the presence of fork DNA, in contrast to ubiquitylation of CMG-Mcm7 (*Figures 2D* and 10 nM [E3]). These findings demonstrate that forked DNA is not a generic inhibitor of SCF$^{Dia2}$, and instead indicate that ubiquitylation of CMG-Mcm7 is specifically impaired by its association with a replication fork, dependent upon the presence of the excluded DNA strand.

## Efficient ubiquitylation of the CMG-replisome is dependent upon recruitment of SCF$^{Dia2}$ by Mrc1 and Ctf4

The ability to ubiquitylate CMG-Mcm7 in a soluble system with defined components made it possible to explore the mechanistic basis for the very high efficiency of CMG ubiquitylation during termination. SCF$^{Dia2}$ was unable to ubiquitylate Mcm7 in reactions in which CMG was replaced by free Cdc45, Mcm2-7 and GINS (*Figure 3A*, compare lanes 1–2). Therefore, Mcm7 ubiquitylation can only take place in the context of the CMG helicase, thereby explaining why Mcm7 ubiquitylation is restricted to S-phase (*Maric et al., 2014*) and cannot occur in the context of Mcm2-7 double hexamers or unloaded Mcm2-7 complexes. Moreover, CMG-Mcm7 ubiquitylation was almost entirely dependent upon the presence of other replisome components (*Figure 3A*, lane 3), indicating that the preferred substrate of SCF$^{Dia2}$ is Mcm7 in the context of the CMG-replisome. Subsequently, a series of dropout experiments indicated that the high efficiency of CMG ubiquitylation was influenced by three specific components of the replisome, namely Pol ε, Mrc1 and Ctf4 (*Figure 3B*). Moreover, CMG-Mcm7 ubiquitylation in the presence of just these three replisome components was equivalent to reactions containing the complete set (*Figure 3—figure supplement 1A–B*).

CMG-Mcm7 ubiquitylation was dramatically impaired in the absence of Ctf4, and ubiquitin chains were substantially shorter in the absence of Mrc1, whereas dropout of Pol ε had a milder effect (*Figure 3B*). Moreover, removal of Pol ε had no impact on CMG ubiquitylation in the absence of Mrc1 (*Figure 3C*, left panel), with which it interacts (*Lou et al., 2008*), suggesting that Pol ε contributes indirectly to CMG ubiquitylation by helping to recruit Mrc1 to the helicase. In contrast, Mrc1 and Ctf4 made additive contributions to CMG ubiquitylation (*Figure 3C*, right panel), and residual CMG ubiquitylation in the absence of both factors was equivalent to reactions containing CMG alone (compare *Figure 3C*, right panel with *Figure 3A*, left panel). Importantly, we confirmed that Mrc1 and Ctf4 were important for efficient CMG ubiquitylation during termination in the reconstituted DNA replication system (*Figure 3D* – ubiquitylation was monitored under conditions where plasmid replication was complete, despite the absence of Mrc1 and Ctf4).

Subsequently, we found that Pol ε, Mrc1 and Ctf4 each contributed to recruitment of SCF$^{Dia2}$ to the CMG helicase (*Figure 3E*, which shows association of the Cdc53 cullin subunit of SCF$^{Dia2}$ with CMG), to an extent that reflected their contribution to CMG-Mcm7 ubiquitylation (*Figure 3F*). These data likely reflect the direct binding of Mrc1 and Ctf4 to the amino-terminal domain of tetratricopeptide repeats (TPR) in Dia2 (*Mimura et al., 2009*; *Morohashi et al., 2009*; *Mukherjee and Labib, 2019*). Note that the impact on CMG ubiquitylation of simultaneously removing both Ctf4 and Mrc1 was not examined in previous in vivo experiments (*Maculins et al., 2015*), since cells lacking both factors are inviable (*Warren et al., 2004*).

In summary, therefore, the inherently high efficiency of CMG ubiquitylation by SCF$^{Dia2}$ reflects a complicated targeting mechanism for the E3 ligase, whereby the replisome components Ctf4 and Mrc1 contribute additively to the recruitment of SCF$^{Dia2}$ to CMG.

## Replisome-coupled ubiquitylation ensures that SCF$^{Dia2}$ pushes CMG beyond a 'ubiquitin threshold' that is inherent to Cdc48-Ufd1-Npl4

The concerted mechanism by which SCF$^{Dia2}$ is recruited to the replisome, via Ctf4 and Mrc1, has two important consequences for CMG ubiquitylation. Firstly, the recruitment mechanism ensures that all CMG complexes are ubiquitylated, once replication fork structure has been lost during termination. In addition, however, replisome tethering of SCF$^{Dia2}$ also predisposes the reaction towards the production of long K48-linked chains on CMG-Mcm7 (*Figures 1–3*). In the absence of Ctf4 and Mrc1, therefore, not only is CMG ubiquitylated less frequently, but the attached chains are shorter (*Figure 3B–C* and *Figure 3—figure supplement 1C*).

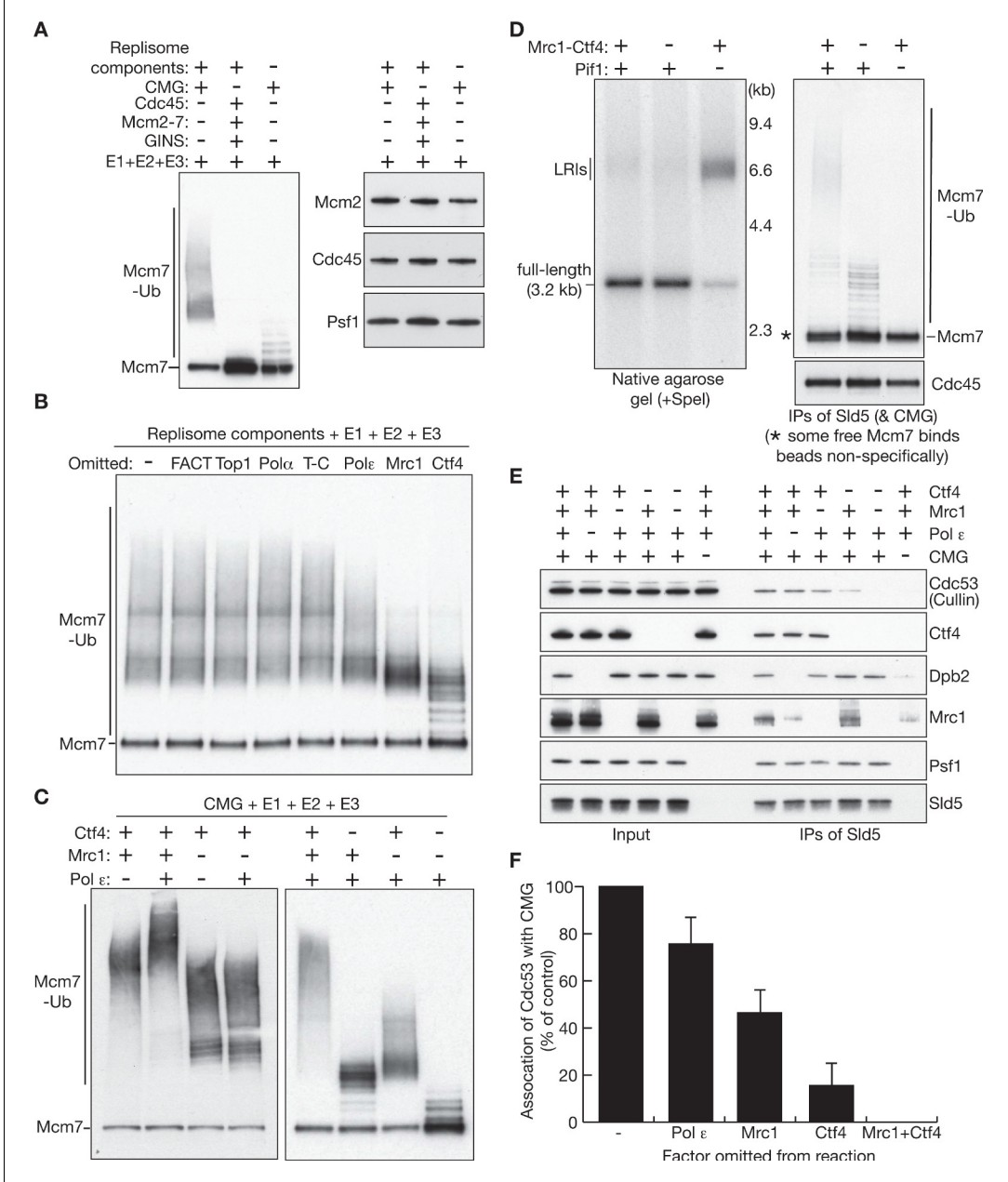

**Figure 3.** The inherently high efficiency of replisome ubiquitylation is dependent upon recruitment of SCF[Dia2] by Mrc1 and Ctf4. (**A**) The indicated factors were incubated at 30°C for 20', and ubiquitylation of Mcm7 was then monitored by immunoblotting, alongside other components of the CMG helicase. (**B**) Analogous reactions were performed in the absence of the indicated replisome components, in order to assess their contribution to CMG-Mcm7 ubiquitylation (T-C = Tof1-Csm3). (**C**) Similar reactions were performed to explore how Ctf4, Mrc1 and Pol ε each contribute to the efficiency of CMG-Mcm7 ubiquitylation. (**D**) Replication-coupled ubiquitylation reactions were performed as in *Figure 1B*, plus or minus the indicated factors. In the absence of Mrc1 and Ctf4, the plasmid template was completely replicated (left panel), but ubiquitylation of CMG-Mcm7 was impaired (right panel). (**E**) The ability of SCF[Dia2] to associate with the CMG helicase was monitored in the presence of replisome components. The indicated factors were mixed, before immunoprecipitation of Sld5 and immunoblotting. Cdc53 = cullin subunit of SCF[Dia2]. (**F**) Quantification of the data in (**E**), to monitor the association of the SCF[Dia2] with the CMG helicase. The experiment was repeated three times, and the figure presents the mean values with standard deviations. See also *Figure 3—figure supplement 1*.

The online version of this article includes the following figure supplement(s) for figure 3:

**Figure supplement 1.** Ctf4 and Mrc1 promote long-chain ubiquitylation of CMG-Mcm7, which leads to efficient CMG disassembly by Cdc48-Ufd1-Npl4.

To explore the functional significance of assembling long ubiquitin chains on CMG-Mcm7, we reconstituted the disassembly of ubiquitylated CMG helicase (*Mukherjee and Labib, 2019*), using recombinant yeast Cdc48-Ufd1-Npl4 (*Figure 4A–B*). At the end of the reaction, the Sld5 subunit of GINS was isolated by immunoprecipitation, in order to monitor its association with Cdc45 and Mcm2-7. In this way, we observed that ubiquitylated CMG was disassembled with extremely high efficiency, dependent upon Cdc48, Ufd1-Npl4 and ubiquitylation (*Figure 4B*, *Figure 4—figure supplement 1A*).

However, CMG disassembly was inefficient when the helicase was ubiquitylated in the absence of Mrc1 and Ctf4 (*Figure 4C*, *Figure 4—figure supplement 1B*). To test whether Mrc1 and Ctf4 were important for the Cdc48-dependent step of disassembly, independently of their role in promoting long-chain CMG ubiquitylation, we split the reaction into several distinct steps (*Figure 4D*; see Materials and methods). Firstly, CMG was ubiquitylated in the presence of Mrc1, which associates dynamically with CMG and supports the conjugation of more than 12 ubiquitins to Mcm7 under these conditions. Secondly, ubiquitylated CMG was isolated by immunoprecipitation of Sld5, and then washed with high salt to remove the associated Mrc1 and SCF$^{Dia2}$. Finally, the reactions were incubated with Cdc48-Ufd1-Npl4. Despite the absence of Ctf4, Mrc1 and SCF$^{Dia2}$ in the final step, ubiquitylated CMG was still disassembled very efficiently by Cdc48-Ufd1-Npl4 (*Figure 4D*). These findings indicated that Mrc1 and Ctf4 promote CMG disassembly by stimulating the formation of long ubiquitin chains on CMG-Mcm7, but are not required subsequently for the action of Cdc48-Ufd1-Npl4.

To investigate the mechanistic implications of ubiquitin chain length for CMG disassembly, we took two approaches that were both dependent on the ability to titrate ubiquitin chain length in the reconstituted in vitro system. Firstly, we explored the relation between the number of ubiquitin moieties conjugated to Mcm7 and the stable recruitment of Cdc48-Ufd1-Npl4 to ubiquitylated CMG, using a Walker B mutant of Cdc48 that does not support substrate unfolding (*Bodnar and Rapoport, 2017*). Surprisingly, stable binding of Cdc48-E588A_Ufd1-Npl4 to ubiquitylated CMG was dependent upon the conjugation of very long polyubiquitin chains to Mcm7 (*Figure 4—figure supplement 1C*), despite previous studies showing that Cdc48-Ufd1-Npl4 can bind with varying affinity to monomeric substrates that are conjugated to three or more ubiquitins (*Bodnar and Rapoport, 2017*), or at least six ubiquitins (*Tsuchiya et al., 2017*). These findings indicate that long ubiquitin chains increase the stability of binding to Cdc48-Ufd1-Npl4, but the data do not address what length of ubiquitin chain is functionally required for a substrate to be unfolded by Cdc48-Ufd1-Npl4.

Secondly, therefore, we developed an assay that provides a direct readout for the number of conjugated ubiquitins that are needed for substrate unfolding. The assay is based on monitoring the ubiquitin chain length of the unfolded Mcm7 subunit that is released upon disassembly of the CMG helicase. Thus, the approach is dependent upon the use of a multimeric ubiquitylated protein complex as the substrate of Cdc48-Ufd1-Npl4, in contrast to previous studies of monomeric ubiquitylated substrates (*Bodnar and Rapoport, 2017*) or free ubiquitin chains (*Tsuchiya et al., 2017*). We established conditions in which a single K48-linked chain of up to about 10 ubiquitin moieties was conjugated to lysine 29 of CMG-Mcm7 (*Figure 4E* and *Figure 4—figure supplements 1D* and 0.3 nM E2, also see Materials and methods), which we previously showed is a favoured site for CMG ubiquitylation in yeast cell extracts (*Maric et al., 2017*). The ability of Cdc48-Ufd1-Npl4 to disassemble CMG with short chains or longer chains was then compared, after isolation of the ubiquitylated helicase by immunoprecipitation of Sld5 (*Figure 4F*). Subsequently, CMG disassembly was monitored by release of ubiquitylated Mcm7 into the supernatant.

As shown in *Figure 4G*, Mcm7 with 1–4 ubiquitins was largely retained on the beads throughout the reaction, indicating that CMG disassembly was blocked. In contrast, Mcm7 with 5–7 ubiquitins was partially released into the supernatant (*Figure 4G*, left side), whereas longer ubiquitin chains on Mcm7 led to rapid dissociation from CMG (*Figure 4G*, right side, 10 nM E2). These findings indicated that CMG disassembly is gated by a 'ubiquitin threshold' that controls the action of Cdc48-Ufd1-Npl4, analogous to an equivalent threshold that is thought to control the degradation of proteasomal substrates (*Swatek and Komander, 2016*). Conjugation of at least five ubiquitins to Mcm7 is required for CMG disassembly, although we note that longer ubiquitin chains increase the efficiency of disassembly still further (consistent with the binding data in *Figure 4—figure supplement 1C*).

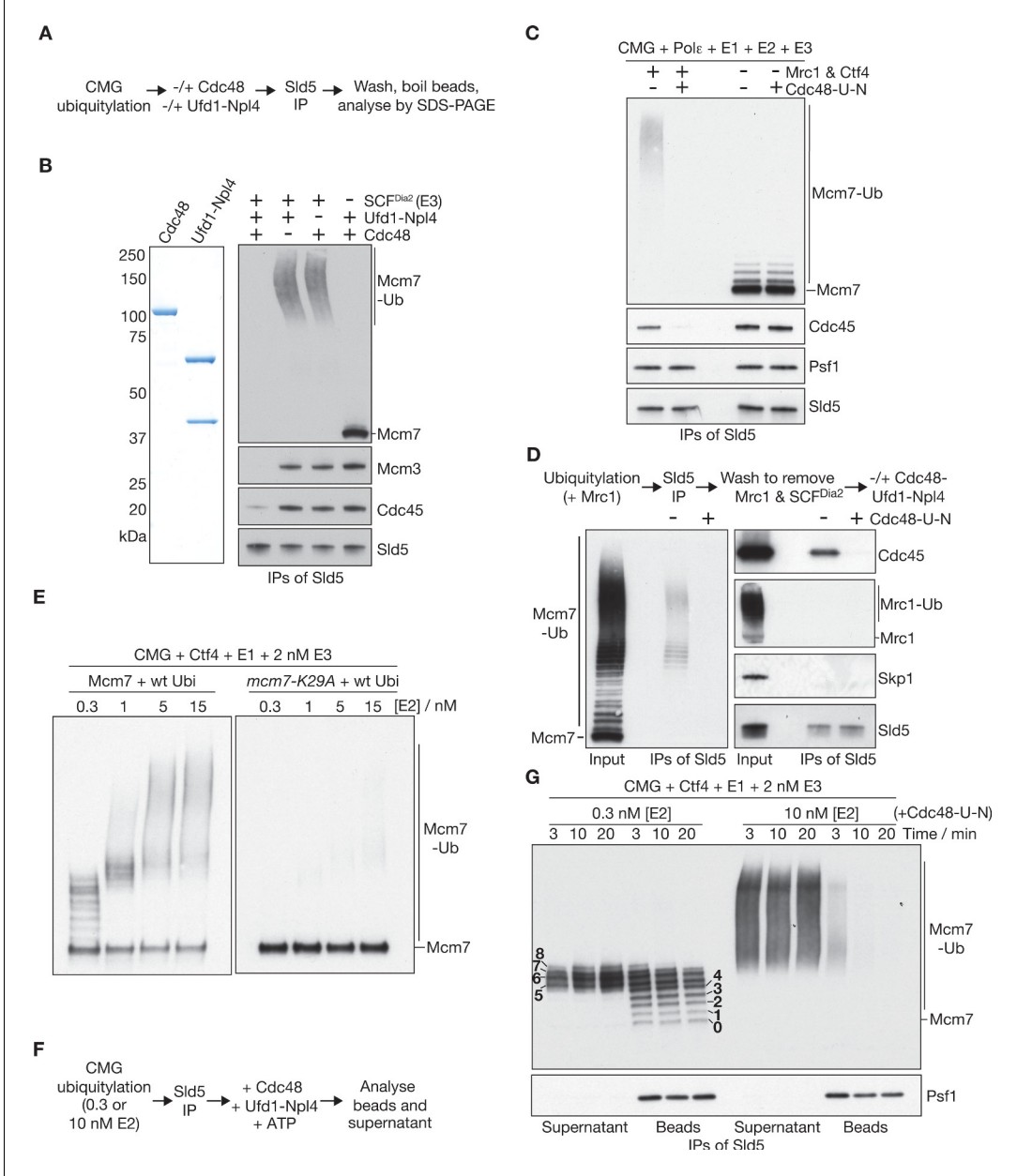

**Figure 4.** Replisome-coupled ubiquitylation ensures that SCF[Dia2] pushes CMG beyond a 'ubiquitin threshold' intrinsic to Cdc48-Ufd1-Npl4. (**A**) Reaction scheme for (**B**). (**B**) Recombinant versions of yeast Cdc48 and Ufd1-Npl4 were purified after expression in bacteria (left panel). Reactions were performed as in (**A**), and the products monitored by immunoblotting (right panels). (**C**) CMG was ubiquitylated in the presence or absence of Mrc1 and Ctf4, in reactions containing 1 nM of E3 (SCF[Dia2]). CMG disassembly by Cdc48-Ufd1-Npl4 was then monitored as above. (**D**) CMG was ubiquitylated in reactions containing Mrc1 and 25 nM SCF[Dia2]. Subsequently, CMG was isolated on anti-Sld5 beads, which were then washed with high salt to remove Mrc1 and SCF[Dia2]. Finally, incubation was continued in the presence or absence of Cdc48-Ufd1-Npl4, and immunoblotting was used to monitor release of the indicated factors from the beads, corresponding to disassembly of CMG. (**E**) Ubiquitylation reactions in the presence of the indicated concentrations of E2, involving CMG with wild-type Mcm7 (left side) or Mcm7-K29A (right side). (**F**) Scheme for disassembly of ubiquitylated CMG bound to beads, as in G. (**G**) Immunoblots for the experiment in (**F**). See also *Figure 4—figure supplement 1*.

The online version of this article includes the following figure supplement(s) for figure 4:

**Figure supplement 1.** CMG helicase disassembly is dependent upon the formation of long K48-linked ubiquitin chains.

In summary, therefore, the replisome components Mrc1 and Ctf4 ensure that SCF$^{Dia2}$ pushes CMG above a ubiquitin threshold that governs the action of Cdc48-Ufd1-Npl4. In this way, the concerted recruitment mechanism for the E3 ligase guarantees that CMG disassembly is highly efficient during DNA replication termination, once the inhibitory effect of DNA replication fork structure has been removed.

## Cdc48-Ufd1-Npl4 selectively unfold the ubiquitylated subunit(s) of CMG to drive replisome disassembly

Cdc48 acts as a 'segregase' that disassembles ubiquitylated protein complexes such as CMG, but previous studies have not determined whether the segregase mechanism involves specific unfolding of the ubiquitylated subunit(s) of such protein complexes, or whether unmodified protomers are also unfolded. This is an important question, since Cdc48/p97 has the ability to unfold both ubiquitylated and unmodified proteins (*Blythe et al., 2017*; *Bodnar and Rapoport, 2017*; *Cheng and Chen, 2015*; *Cooney et al., 2019*; *Wang and Ye, 2018*; *Weith et al., 2018*). The reconstituted CMG disassembly system provides a unique model system with which to address this issue.

We found that CMG disassembly disrupts the association of Mcm2-7 with Cdc45 and the 4-protein GINS complex, but GINS remains intact and can still interact with Cdc45 to some degree (*Figure 5A*, lanes 1–4). Strikingly, ubiquitylated Mcm7 no longer associated with the other 10 CMG subunits after helicase disassembly and instead was bound to Cdc48-Ufd1-Npl4 (*Figure 5A*, lanes 5–6), uniquely amongst the 11 subunits of CMG (*Figures 5A* and *7–8*). To address whether these data reflected the fact that only the ubiquitylated Mcm7 subunit of CMG is unfolded by Cdc48-Ufd1-Npl4, we utilised an assay based on fusion of the bacterial protease FtsH to the carboxyl terminus of Cdc48 (*Bodnar and Rapoport, 2017*), which replaces the AAA+ ATPase that normally supplies unfolded polypeptides to the hexameric FtsH protease (*Figure 5B*).

Degradation by Cdc48-FtsH is dependent upon translocation of peptides through the central channel of Cdc48, thereby providing a proxy for the unfolding of polypeptide substrates by Cdc48-Ufd1-Npl4 (*Bodnar and Rapoport, 2017*). The Cdc48-FtsH fusion protein supported efficient disassembly of ubiquitylated CMG helicase (*Figure 5—figure supplement 1A*), leading to the production of degraded fragments of ubiquitylated Mcm7 (*Figure 5C*). This was dependent not only on Ufd1-Npl4 and ATP (*Figure 5—figure supplement 1B*), but also upon the physical connection of Cdc48 to FtsH (*Figure 5C*, compare lanes 2–3). In contrast, the other 10 subunits of CMG were not degraded by Cdc48-FtsH (*Figure 5D*), except for a very small amount of Mcm4 that correlated with the low level of Mcm4 ubiquitylation in the reconstituted system (*Figure 5D*, *Figure 2—figure supplement 1B–E*, *Figure 5—figure supplement 1B*; note that Mcm4 is adjacent to Mcm7 within the structure of the CMG helicase). These findings demonstrated that Cdc48-Ufd1-Npl4 disassembles ubiquitylated CMG by specifically unfolding the ubiquitylated subunit (almost exclusively Mcm7), which is translocated through the central pore of Cdc48. This then induces the irreversible collapse of the CMG helicase complex.

As noted above, the ubiquitylated Mcm7 subunit of CMG remains associated with Cdc48-Ufd1-Npl4 after helicase disassembly (*Figure 5A*, lanes 7–8). This most likely reflects the fact that the unfolded polypeptide is partially extruded from the C-terminal face of the Cdc48 hexamer, with part of the unfolded protein present within the central channel of Cdc48, and at least some of the K48-linked ubiquitin chain still bound to Ufd1-Npl4 on the N-terminal face of Cdc48 (*Bodnar and Rapoport, 2017*). As observed previously for ubiquitylated GFP (*Bodnar and Rapoport, 2017*), partial cleavage of the remaining ubiquitin chains with the Cdc48-linked deubiquitylase Otu1 (*Ernst et al., 2009*; *Stein et al., 2014*) led to release of unfolded Mcm7 from Cdc48-Ufd1-Npl4 (*Figure 5—figure supplement 1C–D*). In this way, or via the action of other deubiquitylases, the Cdc48 unfoldase is recycled and thus is ready to interact with a new substrate.

## Discussion

Building on previous in vivo approaches and those that used cell extracts, the fully reconstituted assays in this study have enabled us to define the minimal components and molecular basis for the regulation of CMG helicase ubiquitylation, together with the mechanism by which the ubiquitylated CMG is disassembled. As discussed above and summarised in *Figure 6—figure supplement 1*, the data in this study are inconsistent with all of the previously suggested models for the regulation of

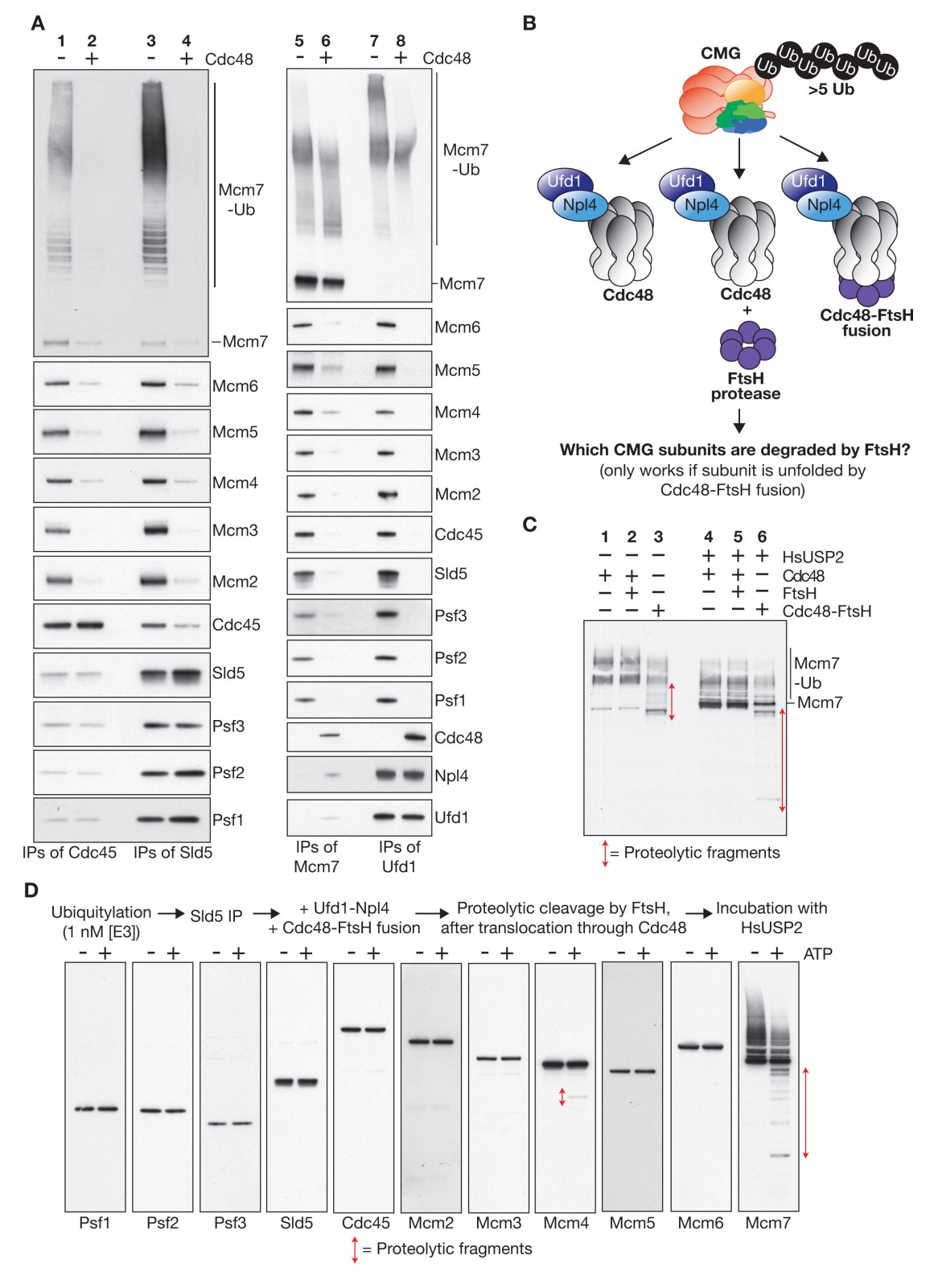

**Figure 5.** Cdc48-Ufd1-Npl4 selectively unfold the ubiquitylated subunit(s) of CMG to drive replisome disassembly. (**A**) CMG was ubiquitylated as in *Figure 4* and then incubated for 20' at 30°C in the presence or absence of Cdc48 as indicated. Ufd1-Npl4 was added to all samples. Subsequently, immunoprecipitations were performed with antibodies to the indicated factors, and the associated factors monitored by immunoblotting. (**B**) Fusion of Cdc48 to the bacterial FtsH protease generates a protein that specifically cleaves unfolded polypeptides that pass through the central channel of the

*Figure 5 continued on next page*

*Figure 5 continued*

Cdc48 hexamer. (**C**) Ubiquitylated CMG was immunoprecipitated with antibodies against Sld5, then incubated with Cdc48 (lanes 1 and 4), Cdc48 + FtsH (lanes 2 and 5) or Cdc48-FtsH fusion protein (lanes 3 and 6), all in the presence of Ufd1-Npl4 and ATP, before treatment for 60' at 30°C with HsUSP2 deubiquitylase (lanes 4–6). Cleaved Mcm7 fragments were then detected by immunoblotting. (**D**) A similar reaction was performed as indicated and all 11 subunits of CMG were monitored by immunoblotting. See also *Figure 5—figure supplement 1*.

The online version of this article includes the following figure supplement(s) for figure 5:

**Figure supplement 1.** Ubiquitylated Mcm7 is unfolded during CMG helicase disassembly, and the ubiquitin chains must then be cleaved in order to release unfolded Mcm7 from Cdc48-Ufd1-Npl4.

CMG ubiquitylation during DNA replication termination. These were each based on a different termination-specific signal that was proposed to be an essential part of the mechanism for CMG ubiquitylation (*Dewar et al., 2015*; *Dewar et al., 2017*; *Maric et al., 2014*; *Sonneville et al., 2017*). Based on our data, we instead propose a new model (*Figure 6*; *Figure 6—figure supplement 1*), whereby ubiquitylation of the CMG-replisome is inherently efficient from the moment of CMG assembly onwards, but ubiquitylation is inhibited throughout elongation by the DNA structure of a replication fork. We note that this model is not dependent on any structural change in CMG itself during DNA replication termination.

According to the new model, the only role of DNA replication termination is to remove the Y-shaped DNA structure of a fork, thereby exposing CMG to ubiquitylation by SCF$^{Dia2}$ and Cdc34. At a pair of converged replication forks, the repressive DNA structure is removed when the two CMG helicases unwind the final base pairs of parental dsDNA and then bypass each other. From this point onwards, each CMG helicase moves along a single parental DNA strand that is no longer connected to the other. Whilst it is likely that CMG encounters the 5' end of the opposing fork's lagging strand and moves onto dsDNA in the few minutes before CMG is disassembled (*Dewar et al., 2015*), the model indicates that the commitment to ubiquitylation occurs when the last bp of parental dsDNA is unwound, and the repressive fork DNA structure is removed from CMG.

Notably, the model also predicts that CMG will be ubiquitylated and disassembled via the same mechanism, whenever replication terminates by a single fork arriving at a telomere or a nick in the leading strand DNA template (*Figure 6*). As above, the trigger is the unwinding of the final bp of dsDNA, after which the repression of CMG ubiquitylation is removed, in this case by the helicase sliding off DNA. For this reason, the ubiquitylation of soluble CMG (*Figures 2–3*) likely represents a physiologically important process that is responsible for disassembling the 32 CMG helicases that are released from the ends of 16 chromosomes in a haploid budding yeast cell, thereby explaining why CMG is not detected after the completion of S-phase (*Gambus et al., 2006*).

We propose that the parental DNA strand that is excluded from the Mcm2-7 channel of CMG, corresponding to the template of the lagging strand at replication forks, sterically inhibits the ubiquitylation of CMG-Mcm7. Consistent with this view, the N-terminal domain of Mcm7 contains a preferred site for ubiquitylation by SCF$^{Dia2}$ and Cdc34 (*Figure 4E* and *Maric et al., 2017*), together with a hairpin that functions as a 'separation pin' to split the two parental DNA strands (*Baretic et al., 2019*). Moreover, replication fork DNA specifically inhibits CMG-Mcm7 ubiquitylation, without affecting the (less efficient) ubiquitylation of other replisome components such as Ctf4 (*Figure 2D*).

In subsequent studies, structural biology will be important to determine how SCF$^{Dia2}$ and Cdc34 engage with the yeast replisome, beyond the interaction of the Dia2-TPR with Ctf4 and Mrc1. In this regard, we note that the C-terminal Leucine-Rich Repeats (LRR) of Dia2, which comprise the canonical substrate-binding domain, (*Cardozo and Pagano, 2004*; *Willems et al., 2004*), are also required for SCF$^{Dia2}$ association with the replisome (*Mukherjee and Labib, 2019*). Given that CMG-Mcm7 is the preferred substrate of SCF$^{Dia2}$ (*Figure 2—figure supplement 1B–E*), we predict that the Dia2-LRR binds directly to CMG. Consistent with this view, CMG is still a target for SCF$^{Dia2}$ in the absence of other replisome components (*Figure 3A*) or the Dia2-TPR (*Maculins et al., 2015*), albeit with much reduced efficiency. Two important future challenges will be to establish the binding site of the Dia2-LRR on CMG and determine the path of the excluded DNA strand at a replication fork. Such structural insights will help to determine whether the excluded DNA strand blocks access of Cdc34 to key lysine residues on the surface of Mcm7, or else interferes directly with the function of SCF$^{Dia2}$,

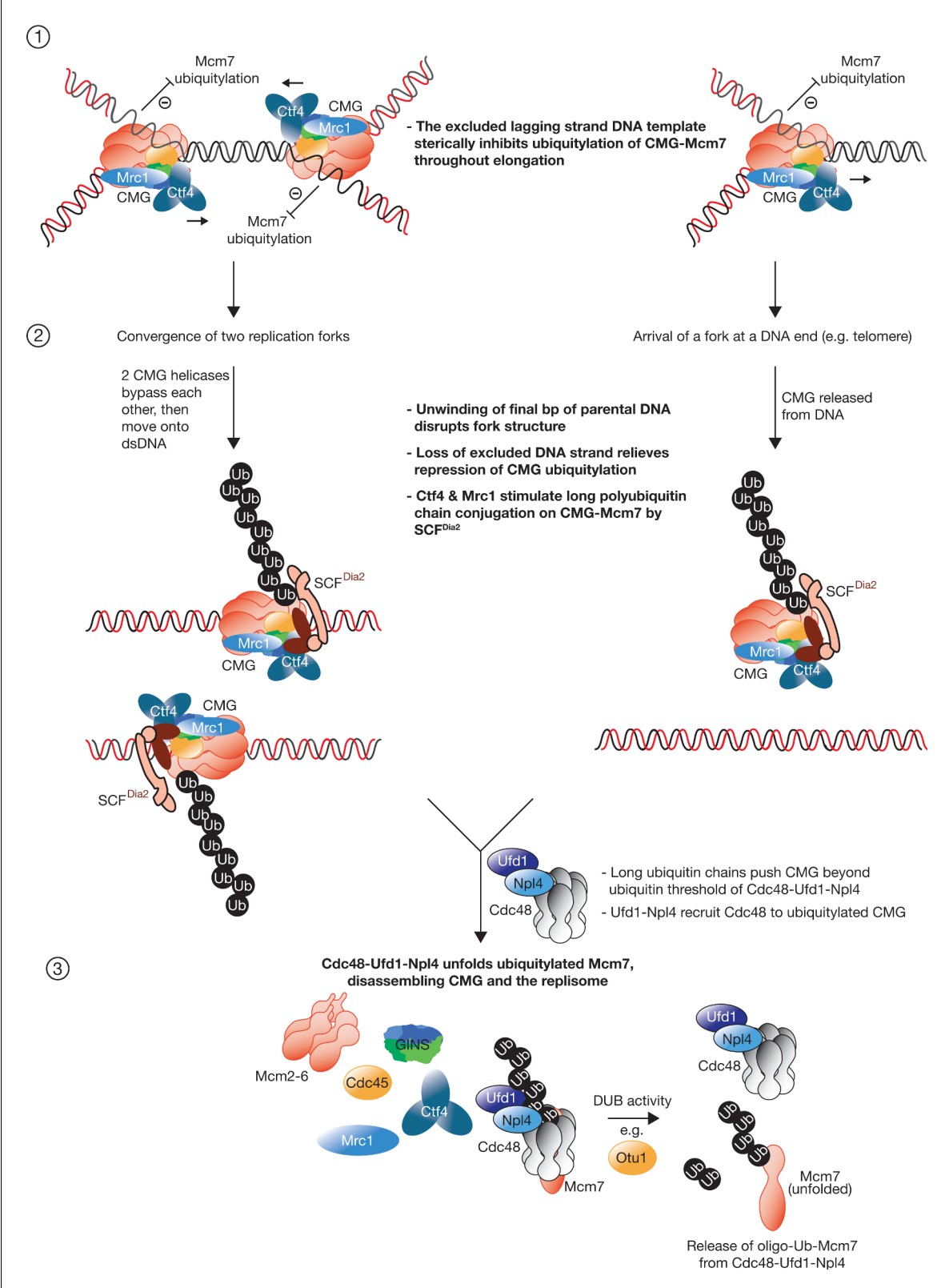

**Figure 6.** Model describing the regulated ubiquitylation and disassembly of the CMG helicase during DNA replication termination. Multiple replisome components are omitted for simplicity. See text for discussion. See also *Figure 6—figure supplement 1*.

The online version of this article includes the following figure supplement(s) for figure 6:

*Figure 6 continued on next page*

*Figure 6 continued*

**Figure supplement 1.** Summary of data that are inconsistent with previous models for the regulation of CMG ubiquitylation and instead support a revised model.

for example by blocking recruitment. In this regard, we note that previous work showed that SCF[Dia2] co-purified with the yeast replisome after DNase treatment of S-phase cell extracts (*Morohashi et al., 2009*), which would have removed the inhibitory effect of the excluded DNA strand. Therefore, it is not known when the ligase is recruited to the yeast replisome during chromosome replication, and this will also be an important issue to address in the future.

Ubiquitylation of nascent CMG is inhibited during initiation in the reconstituted DNA replication system (*Figure 1D–E*), consistent with in vivo observations in yeast cells lacking Mcm10 (*Kanke et al., 2012*; *van Deursen et al., 2012*; *Watase et al., 2012*). This is dependent upon the association of nascent CMG with the DNA template (*Figure 1E* (i)), indicating that the small amount of ssDNA that is associated with CMG under such circumstances (*Douglas et al., 2018*) is sufficient to inhibit the action of SCF[Dia2]. The location of the melted origin DNA in relation to CMG is unclear at present, but it is plausible that ssDNA is associated with the N-terminal tier of Mcm2-7, which contains the 'separation pin' that drives DNA unwinding (*Baretic et al., 2019*). In this case, the mechanism repressing CMG ubiquitylation during initiation and at replication forks might be extremely similar, mediated in each case by unwound parental DNA at the N-terminal face of CMG. However, we cannot rule out that the continued association of firing factors with nascent CMG (*Kanke et al., 2012*) also impairs the action of SCF[Dia2] during initiation.

Binding of Ctf4 and Mrc1 to SCF[Dia2] ensures that a long polyubiquitin chain is conjugated to CMG-Mcm7, once the inhibition of CMG ubiquitylation is relieved upon termination. By studying the release of unfolded Mcm7 from the CMG protein complex, we have established the functional significance of these long polyubiquitin chains, as our data indicate that Cdc48-Ufd1-Npl4 can only unfold proteins that are conjugated to at least five ubiquitins. We note that this functional requirement for at least five ubiquitins during substrate unfolding is consistent with recent data involving cryo-electron microscopy and hydrogen/deuterium exchange mass spectrometry (*Twomey et al., 2019*), indicating that Cdc48-Ufd1-Npl4 can likely engage 4–5 five ubiquitins simultaneously, with two bound by Npl4, 1–2 bound by Ufd1, and a further ubiquitin unfolded within the central channel of Cdc48.

We propose two consequences of this mechanism for the biology of the CMG helicase. Firstly, the requirement for long ubiquitin chains provides a form of quality control. Unscheduled ubiquitylation events during elongation are likely to be inefficient and should thus only produce short chains, which would not trigger premature CMG disassembly by Cdc48-Ufd1-Npl4. Secondly, the functional requirement of Cdc48-Ufd1-Npl4 for long ubiquitin chains has driven the evolution of pathways that exploit short-chain ubiquitylation of CMG during elongation for other uses. For example, work with *Xenopus* egg extracts indicates that the TRAIP ubiquitin ligase leads to slow ubiquitylation of CMG at inter-strand DNA crosslinks, initially favoring the recruitment of DNA repair factors such as the NEIL3 glycosylase to short poly-ubiquitin chains (*Wu et al., 2019*). Our data show why such short-chain ubiquitylation does not trigger premature CMG disassembly. Should the NEIL3 pathway not be sufficient for repair of the lesion, then the subsequent formation of longer ubiquitin chains induces CMG disassembly and the associated Fanconi Anaemia DNA repair pathway (*Räschle et al., 2008*).

Our data demonstrate that the mechanism by which Cdc48-Ufd1-Npl4 disassembles ubiquitylated CMG involves specific unfolding of the ubiquitylated Mcm7 subunit (*Figure 6*, step 3), probably initiated by unfolding of a ubiquitin component of the K48-linked ubiquitin chain that is attached to Mcm7 (*Twomey et al., 2019*). In this way, GINS and Cdc45 are segregated from the remainder of the Mcm2-7 proteins. It is likely that unfolded Mcm7 is then degraded by the proteasome, either directly due to the persistence of an oligo-ubiquitin chain on Mcm7 after release from Cdc48-Ufd1-Npl4, or else via cellular pathways for the ubiquitylation and degradation of misfolded proteins (*Enam et al., 2018*).

These findings provide a general model for the processing of ubiquitylated protein complexes by Cdc48/p97 and its Ufd1-Npl4 cofactors, which is an issue of broad significance in many areas of cell biology (*van den Boom and Meyer, 2018*). Although other adaptors of Cdc48/p97 can recruit the

ATPase to unfold non-ubiquitylated protomers of a protein complex, as is the case for an inhibitor of protein phosphatase 1 (*Weith et al., 2018*), our data indicate that Cdc48-Ufd1-Npl4 exclusively unfolds the ubiquitylated subunit(s) of a protein complex.

Finally, we note that the ubiquitin ligases that control CMG helicase disassembly during DNA replication termination have diverged widely during the course of eukaryotic evolution. Although yeasts utilise SCF$^{Dia2}$ (*Maculins et al., 2015*; *Morohashi et al., 2009*), metazoa employ an unrelated cullin ligase called CUL2$^{LRR1}$ (*Dewar et al., 2017*; *Sonneville et al., 2017*). Nevertheless, the manner by which the CMG helicase embraces its DNA template is likely to be very similar in all eukaryotic species. It will be of particular interest to explore whether the excluded DNA strand defines a common mechanism that constrains CMG ubiquitylation before termination, despite the evolution of different ubiquitin ligases in diverse eukaryotic species.

## Materials and methods

Details of strains and reagents are provided in the Key Resources table in Appendix 1.

### Experimental model and subject details

The *Saccharomyces cerevisiae* strain yJF1 (*MAT**a** ade2-1 ura3-1 his3-11,15 trp1-1 leu2-3,112 can1-100 bar1Δ::hphNT pep4Δ::kanMX*) was transformed with linearised plasmids using standard procedures to generate protein expression strains, as detailed in the Appendix 1-key resources table. For the protein expression strains, the codon usage of the synthetic gene constructs was optimised for high-level expression in *Saccharomyces cerevisiae,* as described previously (*Yeeles et al., 2015*).

For expression of proteins in *E. coli*, plasmids (listed in Appendix 1-key resources table) were transformed into Rosetta (DE3) pLysS cells (Novagen) (F⁻ *ompT hsdS*$_B$(r$_B^-$ m$_B^-$) *gal dcm* (DE3) pLysS-RARE (Cam$^R$)).

### Method details

Protein purification

ORC, Cdc6, Cdt1-Mcm2-7, DDK, S-CDK, Sld3-7, Cdc45, Dpb11, Pol ε, Sld2, GINS, Mcm10, Ctf4, RFC, PCNA, RPA, Mrc1, Pol α - primase, Pif1, Tof1-Csm3 and Top1 were purified based on previously established protocols (*Coster et al., 2014*; *Deegan et al., 2019*; *Frigola et al., 2013*; *On et al., 2014*; *Yeeles et al., 2015*). A brief purification strategy for each of these proteins is listed in the table below. The following proteins were kindly provided by other researchers: Ubiquitin and human USP2b (Dr. Axel Knebel, MRC PPU, Dundee, U.K.); Ulp1 (Dr. Alexander Stein, Max Plank Institute for Biophysical Chemistry, Gottingen, Germany); FACT (Dr. Joe Yeeles, MRC LMB, Cambridge, U.K.); Mcm2-7 (Dr. Max Douglas, Institute of Cancer Research, London, U.K.).

| Protein | Tag | Purification steps |
|---|---|---|
| ORC | CBP-TEV (Orc1) | 1. Calmodulin affinity purification<br>2. HiTrap Q chromatography |
| Cdc6 | GST | 1. GST affinity purification<br>2. Elution by cleavage with 3C protease<br>3. Hydroxyapatite chromatography |
| Cdt1-Mcm2-7 | CBP-TEV (Mcm3) | 1. Calmodulin affinity purification<br>2. Gel filtration (Superdex 200) |
| DDK | CBP-TEV (Dbf4) | 1. Calmodulin affinity purification<br>2. Dephosphorylation (Lambda protein phosphatase)<br>3. Gel filtration (Superdex 200) |
| S-CDK | CBP-TEV (Clb5) | 1. Calmodulin affinity purification<br>2. Elution by cleavage with TEV protease<br>3. Gel filtration (Superdex 200) |
| Sld3-7 | TEV-CBP-PrA (Sld3) | 1. IgG affinity purification<br>2. Elution by cleavage with TEV protease<br>3. Gel filtration (Superdex 200) |

*Continued on next page*

*Continued*

| Protein | Tag | Purification steps |
|---|---|---|
| Cdc45 | Internal 2FLAG | 1. Anti-FLAG affinity purification<br>2. HiTrap Q chromatography |
| Dpb11 | 3FLAG | 1. Anti-Flag affinity purification<br>2. MonoS chromatography |
| Pol ε | TEV-CBP (Dpb4) | 1. Calmodulin affinity purification<br>2. HiTrap heparin chromatography<br>3. Gel filtration (Superdex 200) |
| Sld2 | 3FLAG | 1. Ammonium sulphate precipitation<br>2. Anti-Flag affinity purification<br>3. HiTrap SP chromatography |
| GINS | 6HIS (Psf3) | 1. Ni-NTA affinity purification<br>2. HiTrap Q chromatography<br>3. Gel filtration (Superdex 200) |
| Mcm10 | 6HIS | 1. Ni-NTA affinity purification<br>2. MonoS chromatography (two rounds)<br>3. Gel filtration (Superdex 200) |
| Ctf4 | CBP-TEV | 1. Calmodulin affinity purification<br>2. Gel filtration (Superdex 200) |
| RFC | CBP-TEV (Rfc3) | 1. Calmodulin affinity purification<br>2. HiTrap SP chromatography |
| PCNA | Untagged | 1. Polymin P precipitation of nucleic acids<br>2. Ammonium sulphate precipitation of proteins<br>3. HiTrap SP and HiTrap heparin chromatography (in tandem)<br>4. HiTrap DEAE chromatography<br>5. HiTrap Q chromatography<br>6. Gel filtration (Superdex 200) |
| RPA | CBP-TEV (Rfa1) | 1. Calmodulin affinity purification<br>2. HiTrap heparin chromatography<br>3. HiTrap Q chromatography |
| Mrc1 | 5Flag | 1. FLAG affinity purification<br>2. HiTrap Q chromatography |
| Pol α - primase | CBP-TEV (Pri1) | 1. Calmodulin affinity purification<br>2. Gel filtration (Superdex 200) |
| Pif1 | 6HIS | 1. Ni-NTA affinity purification<br>2. HiTrap SP chromatography<br>3. HiTrap heparin chromatography |
| Tof1-Csm3 | CBP-TEV (Csm3) | 1. Calmodulin affinity purification<br>2. TEV cleavage<br>3. Gel filtration (Superose 6) |
| Top1 | CBP-TEV | 1. Calmodulin affinity purification<br>2. Gel filtration (Superdex 200) |

## Protease inhibitors

One protease inhibitor tablet (Roche, 000000011873580001) was used per 25 ml of lysis buffer where indicated.

1 ml of Sigma protease inhibitor cocktail (Sigma-Aldrich, P8215) was used per 100 ml of lysis buffer where indicated.

## Cdc34

Rosetta cells were transformed with the Cdc34 expression vector (pTDK7). Transformant colonies were inoculated into a 250 ml LB/kanamycin (50 µg/ml)/chloramphenicol (35 µg/ml) culture, which was grown overnight at 37°C with shaking at 200 rpm. The following morning, the culture was diluted into 1 l of LB/kanamycin (50 µg/ml)/chloramphenicol (35 µg/ml) to a final $OD_{600}$ of 0.15. The culture was left to grow at 30°C until an $OD_{600}$ of 1 was reached. 0.8 mM IPTG was added to induce

expression, and cells were incubated for 2.5 hr at 30℃. Cells were harvested by centrifugation at 5000 rpm for 10 min in an JLA-9.1000 rotor (Beckman).

For lysis, cell pellets were resuspended in 20 ml of buffer containing 25 mM Hepes-KOH pH 7.6, 10% glycerol, 0.02% NP-40, 0.5 M NaCl, 20 mM imidazole, Roche protease inhibitor tablets, 1 mM DTT (Cdc34 buffer/20 mM imidazole). Lysozyme was added to a final concentration of 500 µg/ml and the mixture then left for 20 min on ice. Subsequently, the sample was sonicated for 90 s (15 s on, 30 s off) at 40% on a Branson Digital Sonifier. Insoluble material was removed by centrifugation at 15,000 rpm for 30 min in an SS-34 rotor (Sorvall).

The supernatant was subjected to $Ni^{2+}$ affinity purification by incubation with 2 ml packed bead volume of Ni-NTA resin (Qiagen) for 90 min at 4℃. Beads were recovered in a disposable gravity flow column and washed extensively with Cdc34 buffer/20 mM imidazole. Cdc34 was eluted with 16 ml of Cdc34 buffer/0.4 M imidazole without protease inhibitors.

Cdc34-containing fractions were pooled, concentrated to 4 ml, then loaded onto a 120 ml Superdex 75 column in 25 mM Hepes-KOH pH 7.6, 10% glycerol, 0.2 M NaCl, 1 mM DTT. Cdc34-containing fractions were pooled, concentrated, aliquoted and snap froze. About 1.5 mg Cdc34 was purified from a 1 l culture.

## Cdc48

$His_{14}$-SUMO-Cdc48 expressing plasmid was transformed into Rosetta cells. Transformant colonies were inoculated into a 50 ml LB/kanamycin (50 µg/ml)/chloramphenicol (35 µg/ml) culture, which was grown overnight at 37℃ with shaking at 200 rpm. The following morning, each culture was diluted into 1 l of LB/kanamycin (50 µg/ml)/chloramphenicol (35 µg/ml) to a final $OD_{600}$ of 0.15. The culture was left to grow at 30℃ until an $OD_{600}$ of 0.7 was reached. 0.5 mM IPTG was added to induce expression, and cells were incubated for 2 hr at 30℃.

Cells were harvested by centrifugation at 5000 rpm for 10 min in an JLA-9.1000 rotor (Beckman). Bacterial pellets were then resuspended in 20 ml buffer containing 50 mM Tris-Cl pH 8, 0.5 M NaCl, 40 mM imidazole, 5 mM $Mg(OAc)_2$, 0.1 mM ATP, 0.5 mM TCEP, Roche protease inhibitor tablets (Cdc48 buffer/40 mM imidazole). Cells were lysed by addition of lysozyme (1 mg/ml) and 250 U Pierce Universal Nuclease (ThermoFisher Scientific, 88702) followed by incubation at room temperature for 30 min. Insoluble material was removed by centrifugation at 15,000 rpm for 30 min in an SS-34 rotor (Sorvall).

The supernatant was subjected to $Ni^{2+}$ affinity purification by incubation with 1 ml packed bead volume of Ni-NTA resin (Qiagen) for 2 hr at 4℃. Beads were recovered in a disposable gravity flow column and washed extensively with Cdc48 buffer/40 mM imidazole. Cdc48 was eluted with 6 ml of Cdc48 buffer/0.5 M imidazole without protease inhibitors.

Ulp1 protease (10 µg/ml) was added (to cleave the $HIS_{14}$-SUMO tag from Cdc48) and the sample incubated at 4℃ for 30 min. The sample was then concentrated to 500 µl and loaded onto a 24 ml Superose 6 column in 20 mM Hepes-KOH (pH 7.4), 0.3 M sorbitol, 0.15 M NaCl, 5 mM magnesium acetate, 0.1 mM ATP, 0.5 mM TCEP. Cdc48-containing fractions were pooled, concentrated, aliquoted and snap frozen. About 0.5 mg Cdc48 was routinely purified from a 1 l culture.

Cdc48-FtsH was purified as described above for Cdc48, except TB was used in place of LB during bacterial cultures, and protein expression was induced by addition of 0.5 mM IPTG followed by incubation for 16 hr at 18℃.

## CMG

Yeast cells (yTDK20) were grown at 30℃ in YP + 2% raffinose to 2–3 × $10^7$ cells/ml and induced for 3 hr at 30℃ with 2% galactose. 12–15 litres of cells were typically used for each purification. Following expression, cells were collected by centrifugation and washed once with buffer containing 25 mM Hepes KOH pH 7.6, 10% glycerol, 0.02% Tween-20, 2 mM MgOAc, 1 mM DTT, 0.3 M KCl (CMG buffer/0.3 M KCl). The cell pellets were then resuspended in 0.3–0.4 volumes of CMG buffer/0.3 M KCl/Roche protease inhibitor tablets and the resulting suspensions were frozen dropwise in liquid nitrogen. The frozen cells were freezer milled (SPEX CertiPrep 6850 Freezer/Mill) with 4 cycles of 2′ at a rate of 15. The resulting powders were stored at −80℃.

CMG buffer/0.3 M KCl/Roche protease inhibitor tablets was added to thawed powder and the sample was centrifuged (235,000 g, 4°C, 1 hr). The soluble extract was recovered and mixed with 8–10 ml anti-FLAG M2 affinity resin (Sigma) and the mixture incubated at 4°C for 3 hr with rotation.

Resin was collected and washed extensively with CMG buffer/0.3 M KCl/Roche protease inhibitor tablets then CMG buffer/0.3 M KCl without protease inhibitors. CMG was eluted in 1 column volume CMG buffer/0.3 M KCl/0.5 mg/ml 3FLAG peptide, then 1 column volume CMG buffer/0.3 M KCl/0.25 mg/ml 3FLAG peptide.

The eluate fraction was loaded onto a 0.2 ml MiniQ column in CMG buffer/0.3 M KCl. CMG was eluted with a 4 ml gradient from 0.3 to 0.6 M KCl in CMG buffer. CMG containing fractions were then pooled and loaded onto a 24 ml Superose 6 in CMG buffer/0.3 M KCl. Peak fractions containing CMG were pooled and re-loaded onto a 0.2 ml MiniQ column in CMG buffer/0.3 M KCl. CMG was eluted with a 2.5 ml gradient from 0.3 to 0.6 M KCl in CMG buffer.

Peak fractions containing CMG were pooled and dialysed against CMG buffer/0.2 M KOAc at 4°C for 4 hr. The dialysed sample was recovered, aliquoted and snap frozen. A 12 litre culture routinely yielded ~0.3 mg purified CMG.

## FtsH

His$_{14}$-SUMO-FtsH expressing plasmid was transformed into Rosetta cells. A 100 ml LB/kanamycin (50 µg/ml)/chloramphenicol (35 µg/ml) culture was innoculated with transformant colonies and grown overnight at 37°C with shaking at 200 rpm. The following morning, the culture was diluted into 1 l of LB/kanamycin (50 µg/ml)/chloramphenicol (35 µg/ml) to a final OD$_{600}$ of 0.15. The culture was left to grow at 37°C until an OD$_{600}$ of 1 was reached. 0.5 mM IPTG was added to induce expression, and cells were incubated 16 hr at 18°C.

Cells were harvested by centrifugation at 5000 rpm for 10 min in an JLA-9.1000 rotor (Beckman). Bacterial pellets were then resuspended in 20 ml buffer containing 50 mM Tris-Cl pH 8, 0.2 M NaCl, 30 mM imidazole (FtsH buffer/30 mM imidazole). Cells were lysed by addition of lysozyme (0.5 mg/ml) and 250 U Pierce Universal Nuclease (ThermoFisher Scientific, 88702) followed by incubation on ice for 10 min, followed by sonication for 90 s (15 s on, 30 s off) at 40% on a Branson Digital Sonifier. Insoluble material was removed by centrifugation at 15,000 rpm for 30 min in an SS-34 rotor (Sorvall).

The supernatant was subjected to Ni$^{2+}$ affinity purification by incubation with 1 ml packed bead volume of Ni-NTA resin (Qiagen) for 90 min at 4°C. Beads were recovered in a disposable gravity flow column and washed extensively with FtsH buffer/30 mM imidazole. FtsH was eluted with 10 ml of 50 mM Tris-Cl pH 8, 0.1 M NaCl, 0.4 M imidazole, 1 mM TCEP.

Ulp1 protease (10 µg/ml) was added (to cleave the HIS$_{14}$-SUMO tag from FtsH) and the sample incubated on ice overnight. The sample was next loaded onto a 1 ml HiTrap Q HP column pre-equilibrated in 50 mM Tris-Cl pH 8, 0.1 M NaCl. The flow-through was collected and dialysed against 50 mM Tris-Cl pH 7.5, 0.1 M NaCl at 4°C for 3 hr. The dialysed sample was recovered and the HIS-tagged Ulp1 and cleaved HIS-SUMO tag was removed by incubation with 2 ml packed bead volume of Ni-NTA resin (Qiagen) for 1 hr at 4°C. The Ni-NTA flow-through, containing pure FtsH, was collected, aliquoted and snap frozen. The yield of FtsH from 1 l of cells was 9 mg.

## Otu1

Rosetta cells were transformed with the Otu1 expression vector (pTDK35). Transformed colonies were inoculated into a 200 ml LB/kanamycin (50 µg/ml)/chloramphenicol (35 µg/ml) culture, which was grown overnight at 37°C with shaking at 200 rpm. The following morning, the culture was diluted into 2 l of LB/kanamycin (50 µg/ml)/chloramphenicol (35 µg/ml) to a final OD$_{600}$ of 0.15. The culture was left to grow at 30°C until an OD$_{600}$ of 0.4 was reached. 0.5 mM IPTG was added to induce expression, and cells were incubated for 2.5 hr at 30°C. Cells were harvested by centrifugation at 5000 rpm for 10 min in an JLA-9.1000 rotor (Beckman).

For lysis, cell pellets (corresponding to 1 litre of culture) were resuspended in 20 ml of buffer containing 50 mM Tris-Cl pH 7.5, 0.5 M NaCl, 40 mM imidazole, Roche protease inhibitor tablets and 0.5 mM TCEP (Otu1 buffer/40 mM imidazole). Lysozyme was added to a final concentration of 500 µg/ml and the mixture then left for 15 min on ice. Subsequently, the sample was sonicated for 45 s

(15 s on, 30 s off) at 40% on a Branson Digital Sonifier. Insoluble material was removed by centrifugation at 15,000 rpm for 30 min in an SS-34 rotor (Sorvall).

The supernatant was subjected to $Ni^{2+}$ affinity purification by incubation with 1 ml packed bead volume of Ni-NTA resin (Qiagen) for 90 min at 4°C. Beads were recovered in a disposable gravity flow column and washed extensively with Otu1 buffer/40 mM imidazole. Otu1 was eluted with 8 ml of Otu1 buffer/0.5 M imidazole without protease inhibitors.

Ulp1 protease (10 µg/ml) was added (to cleave the $HIS_{14}$-SUMO tag from Otu1) and the sample was dialysed for 16 hr at 4°C vs. 50 mM Tris-Cl pH 7.5, 0.1 M NaCl and then loaded onto a 1 ml HiTrap Q HP column pre-equilibrated in 50 mM Tris-Cl pH 7.5, 0.1 M NaCl. The flow-through was collected, concentrated to 500 µl, then loaded onto a 24 ml Superdex 75 column in 25 mM Hepes-KOH pH 7.6, 0.2 M NaCl, 0.5 mM TCEP. Otu1-containing fractions were pooled, aliquoted and snap frozen. The yield of purified Otu1 from 1 l of cells was ~1.2 mg.

## $SCF^{Dia2}$

Yeast cells (yTDK5) were grown to $2 \times 10^7$ cells/ml then arrested in G1-phase by incubation for 3 hr with 200 ng / ml alpha factor mating pheromone (Pepceuticals). Protein expression was induced for 3 hr at 30°C with 2% galactose. 12 litres of cells were typically used for each purification. Cells were harvested and lysed as described above for purification of CMG, except buffer containing 25 mM Hepes KOH pH 7.6, 10% glycerol, 0.02% NP-40-S, 0.4 M KOAC, 1 mM DTT, Sigma protease inhibitor cocktail and Roche protease inhibitor tablets ($SCF^{Dia2}$ buffer/protease inhibitors) was used in place of CMG buffer.

$SCF^{Dia2}$ buffer/protease inhibitors was added to thawed cell powder and the sample was centrifuged (235,000 g, 4°C, 1 hr). The soluble extract was recovered and mixed with 4 ml IgG Sepharose 6 Fast Flow (GE Healthcare) and the mixture incubated at 4°C for 1 hr with rotation.

Resin was collected and washed extensively with $SCF^{Dia2}$ buffer/protease inhibitors then $SCF^{Dia2}$ buffer without protease inhibitors. Washed beads were resuspended in 8 ml $SCF^{Dia2}$ buffer without protease inhibitors and 400 µg TEV protease was added, followed by incubation at 4°C for 3 hr with rotation.

The TEV eluate fraction was collected, concentrated to 5 ml, then loaded onto a 120 ml Superdex 200 column in $SCF^{Dia2}$ buffer without protease inhibitors. Peak fraction containing $SCF^{Dia2}$ were pooled, concentrated, aliquoted and snap frozen. A 12-l culture routinely yielded ~0.2 mg purified $SCF^{Dia2}$.

## Uba1

HIS-tagged Uba1, expressed in SF21 insect cells and partially purified by Ni-NTA affinity chromatography, was kindly provided by Dr. Axel Knebel. Uba1 was subsequently purified further over a 24 ml Superdex 200 column in 25 mM Hepes-KOH pH 7.6, 0.15 M NaCl, 10% glycerol, 1 mM DTT. Peak fractions containing Uba1 were pooled, aliquoted and snap froze. The yield of purified protein from 1 l of cells was about 22 mg.

## Ufd1-Npl4

Untagged Npl4 and $His_{14}$-SUMO-Ufd1 were expressed separately as follows. Npl4 or $His_{14}$-SUMO-Ufd1 expressing plasmids were transformed into Rosetta cells. Transformant colonies were innoculated into a 50 ml LB/chloramphenicol (35 µg/ml) culture (containing kanamycin (50 µg/ml) for $His_{14}$-SUMO-Ufd1 and ampicillin (50 µg/ml) for Npl4), which was grown overnight at 37°C with shaking at 200 rpm. The following morning, each culture was diluted into 1 l of LB/chloramphenicol (35 µg/ml) (plus kanamycin or ampicillin) to a final $OD_{600}$ of 0.15. The culture was left to grow at 18°C until an $OD_{600}$ of 0.7 was reached. 0.5 mM IPTG was added to induce expression, and cells were incubated for 16 hr at 18°C.

Cells were harvested by centrifugation at 5000 rpm for 10 min in an JLA-9.1000 rotor (Beckman). Bacterial pellets were then mixed at a 6:1 ratio (Npl4:Ufd1) and resuspended in 40 ml buffer containing 50 mM Tris-Cl pH 8, 0.5 M NaCl, 40 mM imidazole, 0.5 mM TCEP, Roche protease inhibitor tablets (UN buffer/40 mM imidazole). Cells were lysed by passing through a C5 Emulsiflex Cell Disruptor (BioPharma Process Systems) three times at 15 kPsi, and insoluble material was removed by centrifugation at 15,000 rpm for 30 min in an SS-34 rotor (Sorvall).

The supernatant was subjected to $Ni^{2+}$ affinity purification by incubation with 1 ml packed bead volume of Ni-NTA resin (Qiagen) for 2 hr at 4°C. Beads were recovered in a disposable gravity flow column and washed extensively with UN buffer/40 mM imidazole. Ufd1-Npl4 was eluted with 8 ml of UN buffer/0.5 M imidazole without protease inhibitors.

Ulp1 protease (10 µg/ml) was added (to cleave the $HIS_{14}$-SUMO tag from Ufd1) and the sample was dialysed for 3 hr at 4°C vs. 50 mM Tris-Cl pH 8, 0.1 M NaCl, 0.5 mM TCEP and then loaded onto a 6 ml Resource Q column pre-equilibrated in 50 mM Tris-Cl pH 8, 0.1 M NaCl. Protein was eluted with a 10-column volume gradient from 0.1 to 0.5 M NaCl. Ufd1-Npl4 containing fractions were pooled, concentrated to 500 µl, then loaded onto a 24 ml Superose 6 column in 20 mM Hepes-KOH (pH 7.4), 0.2 M NaCl, 5 mM magnesium acetate, 0.5 mM TCEP. Ufd1-Npl4-containing fractions were pooled, aliquoted and snap frozen. 1 l of cells yielded 0.6 mg of purified Ufd1-Npl4.

## In vitro replication-ubiquitylation assays

Mcm2-7 loading and DDK phosphorylation was performed by incubating 6 nM 3.2 kb plasmid DNA template (pBS/ARS1WTA), 5–10 nM ORC, 20 nM Cdc6, 40 nM Cdt1/Mcm2-7 and 20 nM DDK in 25 mM Hepes-KOH (pH 7.6), 100 mM potassium acetate, 0.02% NP-40-S, 0.1 mg / ml BSA, 1 mM DTT, 10 mM $Mg(OAc)_2$ and 5 mM ATP at 30°C for 10 min.

Separate buffer and replication protein mixtures were next added sequentially to the Mcm2-7 loading mixture. 10 µl of the Mcm2-7 loading mixture was generally used per sample and this was typically diluted 2-fold in the final reaction. The final replication reaction contained 25 mM Hepes-KOH (pH 7.6), 100 mM potassium acetate, 0.02% NP-40-S, 0.1 mg/ml BSA, 1 mM DTT, 10 mM Mg $(OAc)_2$, 3.75 mM ATP, 30 µM dATP-dCTP-dGTP-dTTP, 33 nM γ-[32P]-dCTP, 400 µM CTP-GTP-UTP, 20 µM creatine phosphate, 50 µg/ml creatine phospho-kinase, 6 µM ubiquitin, 20 nM S-CDK, 30 nM Dpb11, 8 nM GINS, 40 nM Cdc45, 30 nM Pol ε, 5 nM Mcm10, 5 nM RFC, 20 nM PCNA, 20 nM Top1, 20 nM Pol α-primase, 6.25 nM Sld3-7, 40 nM Ctf4, 100 nM RPA, 10 nM Csm3-Tof1, 40 nM Mrc1, 50 nM Sld2 and 5 nM Pif1 (unless otherwise indicated). The extra contribution from protein storage buffers to the final reaction was approximately 22 mM chloride and 50–60 mM acetate, and the corresponding potassium counter-ions. Pol δ was omitted from replication reactions to avoid strand displacement synthesis, which has previously been shown to be enhanced by Pif1 (*Osmundson et al., 2017*; *Rossi et al., 2008*). Top2 is not required for replication termination in the presence of Pif1 (*Deegan et al., 2019*) and was omitted to avoid a slight inhibition of replication efficiency under these conditions.

The replication step was routinely conducted at 30°C for 20 min. For the experiment in *Figure 1B–C*, 30 nM Uba1, 15 nM Cdc34 and 2 nM SCF^Dia2 were added after the replication step, and the incubation continued at 30°C for a further 20 min. For the experiment in *Figure 1D*, Uba1, Cdc34 and SCF^Dia2 were added during the replication step, which was conducted at 30°C for 10 min. Pif1 (5 nM) was then added and the incubation continued at 30°C for a further 20 min. For the experiment in *Figure 1E*, 25 U Pierce Universal Nuclease (ThermoFisher Scientific, 88702) were added after the replication step, before incubation at 30°C for 10 min. Uba1, Cdc34 and SCF^Dia2 were then added and the incubation continued as in *Figure 1B–C*.

Ubiquitylation reactions were stopped by the addition of KOAc to 700 mM. Next, plasmid DNA was digested by addition of 125 U Pierce Universal Nuclease (ThermoFisher Scientific, 88702) and incubation on ice for 30 min. Each sample was then incubated for 30 min at 4°C with 2.5 µl magnetic beads (Dynabeads M-270 Epoxy; 14302D, Life Technologies) that had been coupled to antibodies raised against Sld5. After the incubation, protein complexes bound on antibody-coupled magnetic beads were washed twice with 190 µl of buffer containing 25 mM Hepes-KOH (pH 7.6), 700 mM potassium acetate, 0.02 % NP-40-S, 0.1 mg / ml BSA, 1 mM DTT, 10 mM $Mg(OAc)_2$ (Wash buffer/ 700 mM KOAc). The bound proteins were then eluted by the addition of SDS-PAGE sample loading buffer and boiling for 5 min at 95°C.

For native agarose gel analysis of replication products from these experiments, reactions were quenched by addition of 25 mM EDTA after the ubiquitylation step. SDS (0.1%) and proteinase K (1/ 100 volumes) were subsequently added and the incubation continued at 37°C for 30 min. An equal volume of phenol:chloroform:isoamyl alcohol 25:24:1 (Sigma-Aldrich P2069) saturated with TE (10 mM Tris-HCl pH 8.0, 1 mM EDTA) was next added and the DNA was extracted. Unincorporated nucleotides were removed and the aqueous phase buffer exchanged to TE with Illustra MicroSpin

G-50 columns (GE Healthcare). For restriction digest, 17.75 µl of sample was incubated in 1x CutSmart buffer with 0.25 µl SpeI-HF at 37℃ for 30 min. Digested samples were then separated in 0.8% native agarose gels at 20 V overnight in 1X TAE then dried directly onto chromatography paper (GE Healthcare, 3030–861). The dried gels were typically exposed to both Amersham Hyperfilm ECL (GE Healthcare) and BAS-MS Imaging Plates (Fujifilm), which were then developed on a Typhoon phosphorimager (GE Healthcare).

### In vitro CMG ubiquitylation assays (off DNA)

Reactions (typically 8–10 µl in volume) containing 15 nM CMG, 30 nM Uba1, 15 nM Cdc34, 1 nM SCF$^{Dia2}$, 30 nM Ctf4, 30 nM Pol ε, 45 nM Mrc1, 45 nM FACT, 45 nM Top1, 45 nM Pol α - primase, 45 nM Mcm10 and 45 nM Tof1-Csm3 were assembled on ice in 25 mM Hepes-KOH (pH 7.6), 75 mM potassium acetate, 0.02% NP-40-S, 0.1 mg / ml BSA, 1 mM DTT, 10 mM Mg(OAc)$_2$, 6 µM ubiquitin and 5 mM ATP. From *Figure 3—figure supplement 1B* onwards, FACT, Top1, Pol α - primase, Mcm10 and Tof1-Csm3 were omitted. For Cdc34 titration experiments (*Figure 4E–G* and *Figure 4— figure supplement 1D*), we empirically determined conditions to modify the length of the polyubiquitin chains generated on Mcm7 (Mrc1 and Pol ε were omitted, SCF$^{Dia2}$ was included at 2 nM, and Cdc34 was included at the indicated concentrations). Protein storage buffers typically contributed approximately 50 mM acetate, and the corresponding potassium counter-ions, to the final reaction. Ubiquitylation reactions were conducted at 30℃ for 20 min unless otherwise indicated. Reactions were stopped by the addition of SDS-PAGE sample loading buffer and boiling at 95℃ for 5 min.

### In vitro CMG ubiquitylation assays in the presence of model DNA substrates

Forked DNA substrates were prepared by mixing equimolar amounts of partially complementary oligonucleotides (see Appendix 1-key resources table). The mixture (typically 10 µl in volume, containing each oligonucleotide at a final concentration of 10 µM) was incubated at 95℃ for 10 min followed by gradual cooling to room temperature.

Ubiquitylation reactions performed in the presence of forked DNA were prepared by incubating 15 nM CMG with 50 nM DNA in 25 mM Hepes-KOH (pH 7.6), 75 mM potassium acetate, 0.02% NP-40-S, 0.1 mg / ml BSA, 1 mM DTT, 10 mM Mg(OAc)$_2$, 6 µM ubiquitin and 2 mM ATP for 1 hr on ice. Reactions were shifted to room temperature for 1 min before addition of 30 nM Ctf4, 30 nM Uba1, 15 nM Cdc34 and 1 nM SCF$^{Dia2}$ and incubation at 30℃ for 8 min. Mrc1, Pol ε and other replisome components were omitted from these reactions to limit DNA unwinding by CMG. Reactions were stopped by the addition of SDS-PAGE sample loading buffer and boiling at 95℃ for 5 min.

To monitor DNA binding by CMG in these experiments (as in *Figure 2—figure supplement 2*), reactions (typically 60 µl in volume) containing CMG and Cy3-labeled DNA were assembled as above and incubated on ice for 1 hr. Subsequently, 15 µl magnetic beads (Dynabeads M-270 Epoxy; 14302D, Life Technologies) that had been coupled to antibodies raised against Sld5 were added and the mixture incubated at 4℃ for 1 hr. After the incubation, protein complexes bound on antibody-coupled magnetic beads were washed twice with 190 µl of Wash buffer/150 mM KOAc. For elution, SDS-PAGE sample loading buffer was added to the washed beads followed by boiling for 5 min at 95℃.

### In vitro CMG disassembly assays

For CMG disassembly reactions in solution, CMG ubiquitylation reactions were assembled as above and conducted at 30℃ for 20 min, followed by the addition of 50 nM Cdc48 and 50 nM Ufd1-Npl4, and incubation at 30℃ for a further 20 min. Each sample was then incubated for 1 hr at 4℃ with 2.5 µl magnetic beads (Dynabeads M-270 Epoxy; 14302D, Life Technologies) that had been coupled to antibodies raised against specific CMG subunits as indicated. After the incubation, protein complexes bound on antibody-coupled magnetic beads were washed twice with 190 µl of wash buffer/ 150 mM KOAc. The bound proteins were then eluted by the addition of SDS-PAGE sample loading buffer and boiling for 5 min at 95℃.

For CMG disassembly reactions on beads (*Figure 4D,G*), ubiquitylation reactions (typically 8–10 µl in volume) were conducted at 30℃ for 20 min followed by the addition of potassium acetate to 700 mM. Each sample was then incubated at 4℃ with 2.5 µl magnetic beads (Dynabeads M-270 Epoxy; 14302D, Life Technologies) that had been coupled to anti-Sld5 antibody. After 1 hr, protein

complexes bound to magnetic beads were washed once with 190 µl of Wash buffer/700 mM KOAc then twice with 190 µl of Wash buffer/150 mM KOAc. Beads were resuspended in 10 µl of Wash buffer/150 mM KOAc/5 mM ATP and Cdc48 and Ufd1-Npl4 were added to 50 nM. Reactions were then incubated at 30°C for 20 min with shaking at 1000 rpm. After the incubation, beads and associated proteins were isolated onto a magnetic rack and the supernatant was removed. Beads were then washed twice with 190 µl of Wash buffer/150 mM KOAc. The bound proteins were then eluted by the addition of SDS-PAGE sample loading buffer as above.

For CMG disassembly/proteolysis reactions using Cdc48-FtsH (*Figure 5C–D* and *Figure 5—figure supplement 1B*), reactions were conducted as for regular disassembly reactions on beads, except Cdc48-FtsH was used in place of Cdc48, and the disassembly reaction buffer was supplemented with 20 µM Zn(OAc)$_2$. Where indicated, 400 nM human Usp2b was added to reactions after the Cdc48-FtsH step, and the incubation continued at 30°C for a further 60 min.

For experiments to monitor the release of ubiquitylated Mcm7 from Cdc48-Ufd1-Npl4, dependent on Otu1 (*Figure 5—figure supplement 1C*), reactions were conducted as for regular disassembly reactions in solution, except Otu1 was added after the Cdc48-Ufd1-Npl4 step, and the incubation continued at 30°C for 30 min. Subsequently, each sample was incubated for 1 hr at 4°C with 2.5 µl magnetic beads (Dynabeads M-270 Epoxy; 14302D, Life Technologies) that had been coupled to anti-Ufd1 antibody. After the incubation, protein complexes bound on beads were washed and eluted as for CMG disassembly assays.

## Interaction of SCF$^{Dia2}$ with CMG

Reactions (typically 20 µl in volume) containing 15 nM CMG, 30 nM Uba1, 15 nM Cdc34, 10 nM SCF$^{Dia2}$, 30 nM Ctf4, 30 nM Pol ε and 45 nM Mrc1 were assembled in 25 mM Hepes-KOH (pH 7.6), 75 mM potassium acetate, 0.02% NP-40-S, 0.1 mg / ml BSA, 1 mM DTT, 10 mM Mg(OAc)$_2$, 6 µM ubiquitin and 5 mM ATP and incubated on ice for 10 min. Input samples (typically 5 µl in volume) were removed and the remainder of each sample was then incubated for 30 min at 4°C with 3.75 µl magnetic beads (Dynabeads M-270 Epoxy; 14302D, Life Technologies) that had been coupled to anti-Sld5 antibody. After the incubation, protein complexes bound on beads were washed twice with 190 µl of Wash buffer/150 mM KOAc. The bound proteins were then eluted by the addition of SDS-PAGE sample loading buffer and boiling for 5 min at 95°C.

## Immunoblotting

Protein samples were resolved by SDS–polyacrylamide gel electrophoresis on NuPAGE Novex 4–12% Bis-Tris gels (NP0321 and WG1402A, ThermoFisher Scientific) with NuPAGE MOPS SDS buffer (NP0001, ThermoFisher Scientific), or NuPAGE Novex 3–8% Tris-Acetate gels (EA0375BOX and WG1602BOX, ThermoFisher Scientific) with NuPAGE Tris-Acetate SDS buffer (LA0041, ThermoFisher Scientific). Resolved proteins were either stained with colloidal Coomassie blue dye ('Instant Blue', Expedion), or were transferred onto a nitrocellulose iBlot membrane (Invitrogen) with the iBlot Dry Transfer System (Invitrogen).

Antibodies used for protein detection in this study are described in Appendix 1-key resources table. Conjugates to horseradish peroxidase of anti-sheep IgG from donkey (Sigma, A3415), anti-rabbit IgG from donkey (GE Healthcare, NA934), or anti-goat IgG from rabbit (Sigma, A5420) were used as secondary antibodies before the detection of chemoluminescent signals on Hyperfilm ECL (Amersham, GE Healthcare) using ECL Western Blotting Detection Reagent (GE Healthcare).

## Quantification and statistical analysis

For quantification of immunoblots, ECL Western Blotting Detection Reagent (GE Healthcare) was applied as above, and the membrane then imaged using a ChemiDoc imaging system (Bio-Rad). The Tiff files generated were opened in ImageJ, boxes were drawn around each band, the background signal subtracted, and relative signal calculated as a percentage of the signal in the complete reaction (*Figure 3F*) or reaction lacking Otu1 (*Figure 5—figure supplement 1D*).

The experiments in *Figure 1B–C*, *Figure 2C*, *Figure 3C* (right panel), *Figure 4B*, *Figure 4E*, *Figure 4G*, *Figure 4—figure supplement 1D*, *Figure 5A*, *Figure 5C*, *Figure 5D* and *Figure 5—figure supplement 1B* were each repeated more than three times, as part of many other experiments. For the remaining experiments, the following number of biological replicates were performed:

*Figure 1D* (1x), *Figure 1E* (2x), *Figure 1—figure supplement 2* (1x), *Figure 2B* (3x), *Figure 2D* (1x), *Figure 2E* (3x), *Figure 2F* (1x), *Figure 2—figure supplement 1A* (2x), *Figure 2—figure supplement 1B,D* (3x), *Figure 2—figure supplement 1C,E* (3x), *Figure 2—figure supplement 2* (3x), *Figure 3A* (2x), *Figure 3B* (2x), *Figure 3C* (left panel) (2x), *Figure 3D* (3x), *Figure 3E* (3x), *Figure 3—figure supplement 1B* (1x), *Figure 3—figure supplement 1C* (2x), *Figure 4C* (1x), *Figure 4D* (2x), *Figure 4—figure supplement 1A* (1x), *Figure 4—figure supplement 1B* (1x), *Figure 4—figure supplement 1C* (1x), *Figure 5—figure supplement 1A* (1x), *Figure 5—figure supplement 1C* (3x).

As indicated in the Figure Legends, the mean and standard deviations for the experiments in *Figure 3E–F* and *Figure 5—figure supplement 1C–D* were calculated from three independent experiments.

# Acknowledgements

We gratefully acknowledge the support of the Medical Research Council (core grant MC_UU_12016/13 to KL), the Wellcome Trust (reference 204678/Z/16/Z for a Sir Henry Wellcome Postdoctoral Fellowship to TD and 102943/Z/13/Z for an Investigator award to KL) and Cancer Research UK (Programme Grant C578/A24558 and PhD studentship C578/A25669). We thank Alessandro Costa for yeast strains expressing CMG, Max Douglas for purified Mcm2-7, Michael Jenkyn-Bedford for Cy3-labelled oligonucleotides, Axel Knebel and Clare Johnson for purified Uba1 and Ubiquitin derivatives, Tom Rapoport and Alexander Stein for the plasmids expressing Cdc48-Ufd1-Npl4, Joe Yeeles for purified FACT and MRC PPU Reagents and Services (https://mrcppureagents.dundee.ac.uk) for antibody production. We thank Joe Yeeles and Giulia Saredi for helpful discussions, together with other members of our group.

# Additional information

## Funding

| Funder | Grant reference number | Author |
|---|---|---|
| Medical Research Council | MC_UU_12016/13 | Tom D Deegan<br>Pragya P Mukherjee<br>Ryo Fujisawa<br>Cristian Polo Rivera<br>Karim Labib |
| Wellcome | 102943/Z/13/Z | Karim Labib |
| Wellcome | 204678/Z/16/Z | Tom D Deegan |
| Cancer Research UK | C578/A24558 | Ryo Fujisawa<br>Karim Labib |
| Cancer Research UK | C578/A25669 | Cristian Polo Rivera<br>Karim Labib |

The funders had no role in study design, data collection and interpretation, or the decision to submit the work for publication.

## Author contributions

Tom D Deegan, Conceptualization, Supervision, Funding acquisition, Investigation, Methodology, Writing - review and editing; Progya P Mukherjee, Conceptualization, Investigation, Methodology, PM established conditions for the immunoprecipitation of CMG components, helped to develop the assay for the disassembly of ubiquitylated CMG by Cdc48-Ufd1-Npl4, and performed the original versions of the experiments in Figure 5A and Figure 5-figure supplement 1C; Ryo Fujisawa, Investigation, Methodology, Writing - review and editing, RF discovered the '5-ubiquitin threshold' that controls CMG helicase disassembly by Cdc48-Ufd1-Npl4 (Figure 4G); Cristian Polo Rivera, Validation, Investigation, Methodology, CPR optimised parameters for the in vitro CMG ubiquitylation reactions (Figure 2) and purified CMG-Mcm7-K29A; Karim Labib, Conceptualization, Supervision, Funding acquisition, Writing - original draft, Project administration

**Author ORCIDs**

Karim Labib https://orcid.org/0000-0001-8861-379X

**Decision letter and Author response**

Decision letter https://doi.org/10.7554/eLife.60371.sa1

Author response https://doi.org/10.7554/eLife.60371.sa2

## Additional files

**Supplementary files**

• Transparent reporting form

## Data availability

All data generated or analysed during this study are included in the manuscript and supporting files.

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

# Appendix 1

**Appendix 1—key resources table**

| Reagent type (species) or resource | Designation | Source or reference | Identifiers | Additional information |
|---|---|---|---|---|
| Strain, strain background (*Escherichia coli*) | Rosetta (DE3) pLysS | Novagen | 70956 | N/A |
| Strain, strain background (*Saccharomyces cerevisiae*) | yJF1 | *Frigola et al., 2013* | N/A | Background strain used for construction of yTDK5 |
| Strain, strain background (*Saccharomyces cerevisiae*) | ySDORC | *Frigola et al., 2013* | N/A | ORC purification |
| Strain, strain background (*Saccharomyces cerevisiae*) | yAM33 | *Coster et al., 2014* | N/A | Cdt1-Mcm2-7 purification |
| Strain, strain background (*Saccharomyces cerevisiae*) | ySDK8 | *On et al., 2014* | N/A | DDK purification |
| Strain, strain background (*Saccharomyces cerevisiae*) | yTD6 | *Yeeles et al., 2015* | N/A | Sld3-7 purification |
| Strain, strain background (*Saccharomyces cerevisiae*) | yTD8 | *Yeeles et al., 2015* | N/A | Sld2 purification |
| Strain, strain background (*Saccharomyces cerevisiae*) | yJY13 | *Yeeles et al., 2015* | N/A | Cdc45 purification |
| Strain, strain background (*Saccharomyces cerevisiae*) | yJY23 | *Yeeles et al., 2015* | N/A | Pol α – primase purification |
| Strain, strain background (*Saccharomyces cerevisiae*) | yJY26 | *Yeeles et al., 2015* | N/A | Dpb11 purification |
| Strain, strain background (*Saccharomyces cerevisiae*) | yAJ2 | *Yeeles et al., 2015* | N/A | Polε purification |
| Strain, strain background (*Saccharomyces cerevisiae*) | yAE31 | *Yeeles et al., 2015* | N/A | RPA purification |
| Strain, strain background (*Saccharomyces cerevisiae*) | yAE37 | *Yeeles et al., 2015* | N/A | S-CDK purification |
| Strain, strain background (*Saccharomyces cerevisiae*) | yAE40 | *Yeeles et al., 2015* | N/A | Ctf4 purification |
| Strain, strain background (*Saccharomyces cerevisiae*) | yAE41 | *Yeeles et al., 2015* | N/A | RFC purification |
| Strain, strain background (*Saccharomyces cerevisiae*) | yAE71 | John Diffley | N/A | Mrc1 purification |
| Strain, strain background (*Saccharomyces cerevisiae*) | yTDK4 | *Deegan et al., 2019* | N/A | Csm3-Tof1 purification |
| Strain, strain background (*Saccharomyces cerevisiae*) | yTDK5 | This study | N/A | SCF$^{Dia2}$ purification *MAT**a** ade2-1 ura3-1 his3-11,15 trp1-1 leu2-3,112 can1-100 bar1Δ::hphNT pep4Δ::kanMX ura3::pRS306-SKP1+ ProteinA-3TEV-DIA2 leu2::pRS305-HRT1+CDC53* |
| Strain, strain background (*Saccharomyces cerevisiae*) | yTDK6 | *Deegan et al., 2019* | N/A | Top1 purification |

*Continued on next page*

*Appendix 1—key resources table continued*

| Reagent type (species) or resource | Designation | Source or reference | Identifiers | Additional information |
|---|---|---|---|---|
| Strain, strain background (*Saccharomyces cerevisiae*) | yTDK20 | This study | N/A | CMG purification MATa/MATα pep4Δ::kanMX/pep4Δ::kanMX bar1Δ::hph-NT1/bar1Δ::hph-NT1 ade2−1/ade2-1 ura3−1/ura3-1::pRS306-MCM2-GAL1,10-CBP-TEV-MCM3 his3-11::pRS303-CDC45iFLAG2-GAL1,10-GAL4/his3-11 trp1-1::pRS304-PSF1-GAL1,10-SLD5/trp1-1::pRS304-MCM5-GAL1,10-MCM4 leu2-3::pRS305-PSF2-GAL1,10-PSF3/leu2-3::pRS305-MCM7-GAL1,10-MCM6 ctf4-I901E/ctf4-I901E |
| Strain, strain background (*Saccharomyces cerevisiae*) | yPM224 | This study | N/A | CMG-Mcm7-K29A purification MATa/MATα pep4Δ::kanMX/pep4Δ::kanMX bar1Δ::hph-NT1/bar1Δ::hph-NT1 ade2−1/ade2-1 ura3−1/ura3-1::pRS306-MCM2-GAL1,10-CBP-TEV-MCM3 his3-11::pRS303-CDC45iFLAG2-GAL1,10-GAL4/his3-11 trp1-1::pRS304-PSF1-GAL1,10-SLD5/trp1-1::pRS304-MCM5-GAL1,10-MCM4 leu2-3::pRS305-PSF2-GAL1,10-PSF3/leu2-3::pRS305-MCM7-K29A-GAL1,10-MCM6 ctf4-I901E/ctf4-I901E |
| Antibody | Anti-yeast Mcm2 (sheep polyclonal) | Labib laboratory | 158 | (1:2000) |
| Antibody | Anti-yeast Mcm3 (sheep polyclonal) | Labib laboratory | 16 | (1:1000) |
| Antibody | Anti-yeast Mcm4 (sheep polyclonal) | Labib laboratory | 159 | (1:2000) |
| Antibody | Anti-yeast Mcm5 (sheep polyclonal) | Labib laboratory | 160 | (1:2000) |
| Antibody | Anti-yeast Mcm6 (sheep polyclonal) | Labib laboratory | 161 | (1:2000) |
| Antibody | Anti-yeast Mcm7 (sheep polyclonal) | Labib laboratory | 19 | (1:2000) |
| Antibody | Anti-yeast Psf1 (sheep polyclonal) | Labib laboratory | 58 | (1:2000) |
| Antibody | Anti-yeast Psf2 (sheep polyclonal) | Labib laboratory | 31 | (1:1000) |
| Antibody | Anti-yeast Psf3 (sheep polyclonal) | Labib laboratory | 33 | (1:1000) |
| Antibody | Anti-yeast Sld5 (sheep polyclonal) | Labib laboratory | 32 | (1:1000) |
| Antibody | Anti-yeast Cdc45 (sheep polyclonal) | Labib laboratory | 158 | (1:2000) |

*Continued on next page*

*Appendix 1—key resources table continued*

| Reagent type (species) or resource | Designation | Source or reference | Identifiers | Additional information |
|---|---|---|---|---|
| Antibody | Anti-yeast Ctf4 (sheep polyclonal) | Labib laboratory | 30 | (1:3000) |
| Antibody | Anti-yeast Mrc1 (sheep polyclonal) | Labib laboratory | 125 | (1:1000) |
| Antibody | Anti-yeast Pol2 (sheep polyclonal) | Labib laboratory | 11 | (1:2000) |
| Antibody | Anti-yeast Dpb2 (sheep polyclonal) | Labib laboratory | 122 | (1:2000) |
| Antibody | Anti-yeast Cdc48 (sheep polyclonal) | Labib laboratory | 90 | (1:2000) |
| Antibody | Anti-yeast Ufd1 (sheep polyclonal) | Labib laboratory | 99 | (1:2000) |
| Antibody | Anti-yeast Npl4 (sheep polyclonal) | Labib laboratory | 100 | (1:2000) |
| Antibody | Anti-yeast Cdc53 (rabbit polyclonal) | Santa Cruz Biotechnology | sc-50444 | (1:1000) |
| Antibody | Anti-yeast Skp1 (goat polyclonal) | Santa Cruz Biotechnology | sc-5328 | (1:500) |
| Antibody | Anti-sheep IgG HRP (from donkey) | Sigma-Aldrich | A3415 | (1:10000) |
| Antibody | Anti-rabbit IgG HRP (from donkey) | GE Healthcare | NA934 | (1:10000) |
| Antibody | Anti-goat IgG HRP (from rabbit) | Sigma-Aldrich | A5420 | (1:10000) |
| Recombinant DNA reagent | pAM3 | *Frigola et al., 2013* | N/A | Cdc6 purification |
| Recombinant DNA reagent | pJY19 | *Yeeles et al., 2017* | N/A | PCNA purification |
| Recombinant DNA reagent | pJFDJ5 | *Yeeles et al., 2015* | N/A | GINS purification |
| Recombinant DNA reagent | pET28a-Mcm10 | *Yeeles et al., 2015* | N/A | Mcm10 purification |
| Recombinant DNA reagent | pTF175 | *Biswas et al., 2005* | N/A | FACT purification |
| Recombinant DNA reagent | pJW22 | *Biswas et al., 2005* | N/A | FACT purification |
| Recombinant DNA reagent | pTDK10 | *Deegan et al., 2019* | N/A | Pif1 purification |
| Recombinant DNA reagent | pTDK24 | *Deegan et al., 2019* | N/A | Pif1-K264A purification |
| Recombinant DNA reagent | Ufd1 in K27SUMO | *Stein et al., 2014* | N/A | Ufd1-Npl4 purification |
| Recombinant DNA reagent | Npl4 in pET21b | *Stein et al., 2014* | N/A | Ufd1-Npl4 purification |
| Recombinant DNA reagent | Cdc48 in K27SUMO | *Stein et al., 2014* | N/A | Cdc48 purification |
| Recombinant DNA reagent | Cdc48-FtsH in K27SUMO | *Bodnar and Rapoport, 2017* | N/A | Cdc48-FtsH purification |
| Recombinant DNA reagent | FtsH in K27SUMO | *Bodnar and Rapoport, 2017* | N/A | FtsH purification |
| Recombinant DNA reagent | pTDK3 | This study | N/A | SCF$^{Dia2}$ purification (pRS306-Skp1-Gal1-10-PrA-Dia2) |

*Continued on next page*

*Appendix 1—key resources table continued*

| Reagent type (species) or resource | Designation | Source or reference | Identifiers | Additional information |
|---|---|---|---|---|
| Recombinant DNA reagent | pTDK6 | This study | N/A | SCF$^{Dia2}$ purification (pRS305-Hrt1-Gal1-10-Cdc53) |
| Recombinant DNA reagent | pTDK7 | This study | N/A | Cdc34 purification (Cdc34 in pET28c vector) |
| Recombinant DNA reagent | pTDK35 | This study | N/A | Otu1 purification (Otu1 in K27SUMO vector) |
| Recombinant DNA reagent | pBS/ARS1 WTA | *Marahrens and Stillman, 1992* | N/A | 3.2 kb template for in vitro DNA replication reactions |
| Recombinant DNA reagent | λ HindIII Digest | New England Biolabs | N3012S | Molecular weight marker for agarose gels |
| Sequence-based reagent | 6664 | This study | N/A | *CDC34* forward primer for construction of pTDK7 ATTCTAtctagaaataattttg tttaactttaagaaggagatata ccATGAGTAGTCGCAAA AGCACCGCTTC |
| Sequence-based reagent | 6665 | This study | N/A | *CDC34* reverse primer for construction of pTDK7 atcgatCTCGAGtgatccgc cctgaaaatacaggttttcTA TTTTCTTTGAAACTC TTTCTACATCCTC |
| Sequence-based reagent | 8302 | This study | N/A | *OTU1* forward primer for construction of pTDK35 gaacagattggtggcATGA AACTGAAAGTTAC TGGAGCAGG |
| Sequence-based reagent | 8303 | This study | N/A | *OTU1* reverse primer for construction of pTDK35 gtgcggccgcttattaTCTA TTTTGGCCAAAATCAACG |
| Sequence-based reagent | Unblocked leading | This study | N/A | Leading strand template for construction of model replication fork DNA TAGAGTAGGAAGTGATG GTAAGTGATTAGAGAATT GGAGAGTGTGTTTTTTTT TTTTTTTTTTTTTTTTTT TTTTTTTT*T*T*T*T*T [* denotes a phosphorothioate bond] |
| Sequence-based reagent | Biotinylated lagging (15 nt arm) | This study | N/A | Lagging strand template for construction of model replication fork DNA (15 nt 5' flap) GGCAGGCAGGCAGGCA CACACTCTCCAATTCTCT AATCACTTACCATCACT TCCTACTCTA- DesthioBiotin-TEG |

*Continued on next page*

*Appendix 1—key resources table continued*

| Reagent type (species) or resource | Designation | Source or reference | Identifiers | Additional information |
|---|---|---|---|---|
| Sequence-based reagent | Biotinylated lagging (five nt arm) | This study | N/A | Lagging strand template for construction of model replication fork DNA (5 nt 5' flap) CAGGCACACACTCTCCAA TTCTCTAATCACTTACCA TCACTTCCTACTCTA-DesthioBiotin-TEG |
| Sequence-based reagent | Biotinylated lagging (no arm) | This study | N/A | Lagging strand template for construction of model replication fork DNA (no 5' flap) ACACACTCTCCAATTCT CTAATCACTTACCATCA CTTCCTACTCTA-DesthioBiotin-TEG |
| Sequence-based reagent | DBO2 | Joe Yeeles | N/A | Cy3-labelled leading strand template for construction of model replication fork DNA Cy3- TAGAGTAGGAAGTGA TGG TAAGTGATTAGAGAATTG GAGAGTGTGTTTTTTTTTT TTTTTTTTTTTTTTTTTTT TTTTTT*T*T*T*T*T [* denotes a phosphorothiote bond, T is internally biotinylated] |
| Peptide, recombinant protein | ORC | *Frigola et al., 2013* | N/A | N/A |
| Peptide, recombinant protein | Cdc6 | *Frigola et al., 2013* | N/A | N/A |
| Peptide, recombinant protein | Cdt1- Mcm2-7 | *Coster et al., 2014* | N/A | N/A |
| Peptide, recombinant protein | Mcm2-7 | Max Douglas | N/A | N/A |
| Peptide, recombinant protein | DDK | *On et al., 2014* | N/A | N/A |
| Peptide, recombinant protein | S-CDK | *Yeeles et al., 2015* | N/A | N/A |
| Peptide, recombinant protein | Sld3-7 | *Yeeles et al., 2015* | N/A | N/A |
| Peptide, recombinant protein | Cdc45 | *Yeeles et al., 2015* | N/A | N/A |
| Peptide, recombinant protein | Dpb11 | *Yeeles et al., 2015* | N/A | N/A |
| Peptide, recombinant protein | Sld2 | *Yeeles et al., 2015* | N/A | N/A |
| Peptide, recombinant protein | Pol ε | *Yeeles et al., 2015* | N/A | N/A |
| Peptide, recombinant protein | GINS | *Yeeles et al., 2015* | N/A | N/A |
| Peptide, recombinant protein | Mcm10 | *Yeeles et al., 2015* | N/A | N/A |
| Peptide, recombinant protein | Pol α - primase | *Yeeles et al., 2015* | N/A | N/A |

*Appendix 1—key resources table continued*

| Reagent type (species) or resource | Designation | Source or reference | Identifiers | Additional information |
|---|---|---|---|---|
| Peptide, recombinant protein | RPA | *Yeeles et al., 2015* | N/A | N/A |
| Peptide, recombinant protein | Ctf4 | *Yeeles et al., 2015* | N/A | N/A |
| Peptide, recombinant protein | Mrc1 | *Yeeles et al., 2017* | N/A | N/A |
| Peptide, recombinant protein | Csm3-Tof1 | *Deegan et al., 2019* | N/A | N/A |
| Peptide, recombinant protein | RFC | *Yeeles et al., 2017* | N/A | N/A |
| Peptide, recombinant protein | PCNA | *Yeeles et al., 2017* | N/A | N/A |
| Peptide, recombinant protein | Top1 | *Deegan et al., 2019* | N/A | N/A |
| Peptide, recombinant protein | FACT | Joe Yeeles | N/A | N/A |
| Peptide, recombinant protein | Pif1 | *Deegan et al., 2019* | N/A | N/A |
| Peptide, recombinant protein | Pif1-K264A | *Deegan et al., 2019* | N/A | N/A |
| Peptide, recombinant protein | CMG | This study | N/A | Details in Material and Methods |
| Peptide, recombinant protein | CMG-Mcm7-K29A | This study | N/A | Details in Material and Methods |
| Peptide, recombinant protein | Uba1 | This study | N/A | Details in Material and Methods |
| Peptide, recombinant protein | Cdc34 | This study | N/A | Details in Material and Methods |
| Peptide, recombinant protein | SCF$^{Dia2}$ | This study | N/A | Details in Material and Methods |
| Peptide, recombinant protein | Ubiquitin | Axel Knebel | N/A | N/A |
| Peptide, recombinant protein | USP2b | Axel Knebel | N/A | N/A |
| Peptide, recombinant protein | Ulp1 | Alexander Stein | N/A | N/A |
| Peptide, recombinant protein | Ufd1-Npl4 | *Stein et al., 2014* | N/A | N/A |
| Peptide, recombinant protein | Cdc48 | *Stein et al., 2014* | N/A | N/A |
| Peptide, recombinant protein | Cdc48-FtsH | *Bodnar and Rapoport, 2017* | N/A | N/A |
| Peptide, recombinant protein | FtsH | *Bodnar and Rapoport, 2017* | N/A | N/A |
| Software, algorithm | ImageJ | National Institute of Health | https://imagej.nih.gov/ij/ | N/A |

