## [Decision Letter]

**Acceptance summary:**

Recent studies on eukaryotic DNA replication have shown that it is possible to reconstitute the entire replication process with purified proteins and the current paper extends this to reconstitute termination of DNA replication and removal of the replicative DNA helicase from the replicated DNA. In so doing, the authors have discovered a new mechanism for linking ubiquitylation of the helicase subunit, Mcm7, to the structure of the replicated DNA. This is an important contribution to understanding how the genome is copied only once per cell division cycle.

**Decision letter after peer review:**

Thank you for submitting your article "CMG helicase disassembly is controlled by replication fork DNA, replisome components and a ubiquitin threshold" for consideration by *eLife*. Your article has been reviewed by three peer reviewers, including Bruce Stillman as the Reviewing Editor and Reviewer #1, and the evaluation has been overseen by Jessica Tyler as the Senior Editor. The following individuals involved in review of your submission have agreed to reveal their identity: Dirk Remus (Reviewer #2); Philip Zegerman (Reviewer #3).

The reviewers have discussed the reviews with one another and the Reviewing Editor has drafted this decision to help you prepare a revised submission.

Summary:

This paper describes the reconstitution of replisome disassembly after the termination of DNA replication with purified budding yeast proteins. Replisome disassembly has been previously shown by the Labib lab and others to be mediated by the ubiquitylation-dependent extraction of Mcm7 by the Cdc48 ATPase. However, the mechanism that links replisome disassembly to the termination of DNA replication has been elusive. Here, the authors propose that the displaced lagging strand template inhibits replisome disassembly during replication elongation. Moreover, the authors propose that replisome disassembly requires conjugation of extended ubiquitin chains on Mcm7 as a failsafe mechanism against premature replisome disassembly due to potential stochastic Mcm7 ubiquitylation. In addition, the authors demonstrate novel roles for Ctf4 and Mrc1 in mediating Mcm7 ubiquitylation.

This is a very interesting study that addresses an important biological problem and provides several critical mechanistic insights into an essential step of the DNA replication reaction. The experiments are generally well-controlled and of high quality. There are a number of minor clarifications and comments that the authors should address in a revised manuscript, none requiring new data.

Minor points:

1) The purification of Cdc34, Cdc48, FtsH, Otu1, SCFDAI2, Uba1, Ufd1-Npl4, and CMG are described in the Materials and methods. The authors should state the yield of purified protein from the amount of cells used.

2) Discussion: The authors discuss why Mcm7 in the CMG is not ubiquitylated when it is an active helicase and the model is fine and consistent with the data. Perhaps they should also discuss why Mcm7 in all of the loaded Mcm2-7 double hexamers (DH) is not ubiquitylated. Two possibilities come to mind, one is the lack of elongation factors associated with the DH (e.g., Ctf4, Mrc1) or that the juxtaposition of the DH Mcm2-7 amino termini buries the Mcm7 N-terminal tails, making then only accessible after DH dissociation. The structures of the Mcm2-7 DH show some accessibility of the Mcm7 amino terminus, so it is either masked or elongation factors are required.

3) The paper relies on a system that utilizes Pif1 to induce replication termination in reactions containing Top1 but lacking Top2. This may not represent the default termination pathway. For example, the authors had previously demonstrated that termination of plasmid replication in yeast cells is only slightly delayed after depletion of Pif1/Rrm3 (PMID 30850330), indicating that termination is not strictly dependent on Pif1/Rrm3 in vivo. Moreover, as demonstrated in PMID 32341532 and the authors' previous paper (PMID 30850330) termination is less efficient in the presence of Top1 compared to Top2. Therefore: Why are the reactions here performed in the absence of Top2? Moreover, termination efficiency in the reconstituted budding yeast system reported in PMID 32341532 is significantly higher than that reported by the authors' system here, indicating that specific reaction conditions significantly affect the termination process. Combined, it therefore seems relevant to point out at the beginning of the paper that the Top1/Pif1 system is being employed here as a useful tool to enrich for LRIs.

4) Could the authors explain why the replication experiments are performed in the absence Pol-δ, which is required for normal DNA replication?

5) Figure 4: The authors had previously shown that K29 ubiquitylation is not essential for replisome disassembly in vivo (PMID 28355556). Thus, while the specificity for K29 ubiquitylation observed in the in vitro system here is a useful tool to assess the ubiquitin chain length requirement for Cdc48-catalyzed replisome disassembly (Figure 4), it should be noted that alternative disassembly pathways remain possible.

---

## [Author Response]

Minor points:1) The purification of Cdc34, Cdc48, FtsH, Otu1, SCFDAI2, Uba1, Ufd1-Npl4, and CMG are described in the Materials and methods. The authors should state the yield of purified protein from the amount of cells used.

For each of the purified proteins mentioned by the reviewers, the yield has now been included at the end of the corresponding section of Materials and methods.

2) Discussion: The authors discuss why Mcm7 in the CMG is not ubiquitylated when it is an active helicase and the model is fine and consistent with the data. Perhaps they should also discuss why Mcm7 in all of the loaded Mcm2-7 double hexamers (DH) is not ubiquitylated. Two possibilities come to mind, one is the lack of elongation factors associated with the DH (e.g., Ctf4, Mrc1) or that the juxtaposition of the DH Mcm2-7 amino termini buries the Mcm7 N-terminal tails, making then only accessible after DH dissociation. The structures of the Mcm2-7 DH show some accessibility of the Mcm7 amino terminus, so it is either masked or elongation factors are required.

We previously showed that Mcm2-7 complexes are only ubiquitylated during S-phase in yeast cells, dependent upon Cdc45 (Maric et al., 2014). This indicated that only CMG complexes and not Mcm2-7 double hexamers are a substrate for SCF^Dia2^ (especially since we had also shown in the 2014 paper that CMG ubiquitylation can occur in G1-phase, upon re-expression of Dia2 in G1-arrested cells that had previously been depleted for Dia2). As the reviewers note, the failure to ubiquitylate Mcm2-7 double hexamers could either be due to a requirement for elongation factors, or else might reflect the occlusion within the double hexamer interface of important binding sites for SCF^Dia2^.

However, our new data in the present manuscript show that free Mcm2-7 complexes (single hexamers) are not ubiquitylated by SCF^Dia2^ (Figure 3A lane 2). Therefore, the failure to ubiquitylate Mcm2-7 double hexamers is unlikely to be due to the occlusion of important binding sites for SCF^Dia2^ within the double hexamer interface. Instead, only CMG is a substrate for SCF^Dia2^ and ubiquitylation is almost entirely dependent upon the Ctf4 and Mrc1 partners of CMG (compare lanes 1-3 of Figure 3A).

In the revised manuscript we now discuss why Mcm2-7 complexes cannot be ubiquitylated outside of S-phase (subsection “Efficient ubiquitylation of the CMG-replisome is dependent upon recruitment of SCF^Dia2^ by Mrc1 and Ctf4”).

3) The paper relies on a system that utilizes Pif1 to induce replication termination in reactions containing Top1 but lacking Top2. This may not represent the default termination pathway. For example, the authors had previously demonstrated that termination of plasmid replication in yeast cells is only slightly delayed after depletion of Pif1/Rrm3 (PMID 30850330), indicating that termination is not strictly dependent on Pif1/Rrm3 in vivo. Moreover, as demonstrated in PMID 32341532 and the authors' previous paper (PMID 30850330) termination is less efficient in the presence of Top1 compared to Top2. Therefore: Why are the reactions here performed in the absence of Top2? Moreover, termination efficiency in the reconstituted budding yeast system reported in PMID 32341532 is significantly higher than that reported by the authors' system here, indicating that specific reaction conditions significantly affect the termination process. Combined, it therefore seems relevant to point out at the beginning of the paper that the Top1/Pif1 system is being employed here as a useful tool to enrich for LRIs.

It is true, as the reviewers note, that we use Pif1-dependent DNA replication termination as a useful tool to compare ubiquitylation in the absence (-Pif1) or presence (+Pif1) of termination. We point this out at the beginning of the Results: “replication reactions in the presence or absence of Pif1 provide a model system, with which to study the regulation of CMG ubiquitylation during DNA replication termination”.

However, our choice of topoisomerase was not related to the efficiency of termination. The reviewers correctly note that termination is slightly more efficient in our system in the presence of Top2, compared to reactions that only have Top1 (Deegan et al., 2019). But even in the presence of Top2, more than 80% of replicated plasmids accumulate as ‘late replication intermediates’, due to a failure of DNA replication termination.

Furthermore, Top2 is not essential for replication termination either in vivo (Baxter et al., 2008) or in vitro (Deegan et al., 2019). For this study, we omitted Top2 in order to avoid a slight inhibitory effect of Top2 on replication efficiency under our reaction conditions. We have now added a note to this effect in Materials and methods (subsection “In vitro replication-ubiquitylation assays”).

4) Could the authors explain why the replication experiments are performed in the absence Pol-δ, which is required for normal DNA replication?

Pol-δ was omitted to avoid strand displacement synthesis, which is driven by Pol-δ and was previously shown to be enhanced by Pif1 (Osmundson et al., 2017, Rossi et al., 2008). Although this strand displacement activity of Pol-δ can be alleviated by higher salt concentrations, the latter conditions would also impair the efficiency of ubquitylation. We now state this and cite the relevant papers in Materials and methods (subsection “In vitro replication-ubiquitylation assays”).

5) Figure 4: The authors had previously shown that K29 ubiquitylation is not essential for replisome disassembly in vivo (PMID 28355556). Thus, while the specificity for K29 ubiquitylation observed in the in vitro system here is a useful tool to assess the ubiquitin chain length requirement for Cdc48-catalyzed replisome disassembly (Figure 4), it should be noted that alternative disassembly pathways remain possible.

Our previous work showed that Mcm7-K29 is the only residue that is ubiquitylated by SCF^Dia2^-Cdc34 during in vitro ubiquitylation reactions in yeast cell extracts, under conditions that are relatively inefficient (Maric et al., 2017). Mass spectrometry analysis confirmed K29 as a bona fide ubiquitylation site in vivo, but we found that other sites on Mcm7 could also be ubiquitylated in yeast cells with high efficiency (Maric et al., 2017). Consistent with these findings, our new data in the present manuscript show that multiple sites on Mcm7 can be ubiquitylated in the reconstituted ubiquitylation system under optimal conditions (e.g. see the K0-ubiquitin samples in Figure 2—figure supplement 1C with 25 nM E3 and 15 nM E2). However, Mcm7-K29 is the only site that is modified if the efficiency of the reactions is reduced (e.g. Figure 4E with 1 nM E3 and 0.3 nM E2). We now state that more than one lysine on Mcm7 is modified under standard reaction conditions in the subsection “CMG ubiquitylation is inherently efficient in the presence of other replisome proteins”.

Together with our previous in vivo findings (Maric et al., 2017), these data indicate that ubiquitylation of other sites on Mcm7, in addition to Mcm7-K29, can also support CMG disassembly. However, there is currently no reason to suggest the existence of alternative disassembly pathways that are independent of the ubiquitylation of CMG-Mcm7 (at least in budding yeast).